



# Four-dimensional temperature, salinity and mixed layer depth in the Gulf Stream, reconstructed from remote sensing and in situ observations with neural networks.

Etienne Pauthenet[1], Loïc Bachelot[2], Kevin Balem[1], Guillaume Maze[1], Anne-Marie Tréguier[1], Fabien Roquet[3], Ronan Fablet[4], and Pierre Tandeo[4]

[1]Ifremer, Univ. Brest, CNRS, IRD, Laboratoire d'Océanographie Physique et Spatiale (LOPS), IUEM, 29280, Plouzané, France.
[2]Ifremer, Univ. Brest, CNRS, IRD, Service Ingénierie des Systèmes d'Information (PDG-IRSI-ISI), IUEM, 29280, Plouzané, France.
[3]Department of Marine Sciences, University of Gothenburg, Gothenburg, Sweden.
[4]IMT Atlantique, CNRS UMR Lab-STICC, Brest, France.

**Correspondence:** Etienne Pauthenet (etienne.pauthenet@ifremer.fr)

**Abstract.** Despite the ever-growing amount of ocean's data, the interior of the ocean remains under sampled in regions of high variability such as the Gulf Stream. In this context, neural networks have been shown to be effective for interpolating properties and understanding ocean processes. We introduce OSnet (Ocean Stratification network), a new ocean reconstruction system aimed at providing a physically consistent analysis of the upper ocean stratification. The proposed scheme is a bootstrapped

multilayer perceptron trained to predict simultaneously temperature and salinity (T-S) profiles down to 1000m and the Mixed Layer Depth (MLD) from surface data covering 1993 to 2019. OSnet is trained to fit sea surface temperature and sea level anomalies onto all historical in-situ profiles in the Gulf Stream region. To achieve vertical coherence of the profiles, the MLD prediction is used to adjust a posteriori the vertical gradients of predicted T-S profiles, thus increasing the accuracy of the solution and removing vertical density inversions. The prediction is generalized on a 1/4° daily grid, producing four-dimensional

fields of temperature and salinity, with their associated confidence interval issued from the bootstrap. OSnet profiles have root mean square error comparable with the observation-based Armor3D weekly product and the physics-based ocean reanalysis Glorys12. The maximum of uncertainty is located north of the Gulf Stream, between the shelf and the current, where the thermohaline variability is large. The OSnet reconstructed field is coherent even in the pre-ARGO years, demonstrating the good generalization properties of the network. It reproduces the warming trend of surface temperature, the seasonal cycle of

surface salinity and mesoscale structures of temperature, salinity and MLD. While OSnet delivers an accurate interpolation of the ocean's stratification, it is also a tool to study how the interior of the ocean's behaviour reflects on surface data. We can compute the relative importance of each input for each T-S prediction and analyse how the network learns which surface feature influences most which property and at which depth. Our results are promising and demonstrate the power of machine learning methods to improve the prediction of ocean interior properties from observations of the ocean surface.



## 1 Introduction

*In situ* observations of the ocean's vertical structure are accurate but sparsely distributed in time and space, hampering the study of mesoscale features (Siegelman et al., 2020a) and the computation of large scale integrated variables such as ocean heat content (Wang et al., 2018; Durack et al., 2014). Meanwhile the ocean's surface is observed at high temporal and spatial resolution with satellites since the early 1990s. Remote sensing allows to observe surface signature of mesoscale to subme-soscale features (Siegelman et al., 2020b), and to track climatic trends of sea surface height (Nerem et al., 2018), temperature (Merchant et al., 2019) and salinity (Reul et al., 2020). It is therefore highly valuable to combine sparse *in situ* profiles and high resolution remote sensing observations in order to predict the ocean's stratification at higher resolution and frequency.

This problem can be approached from two main points of view. First, the physical approach aims at constraining a global circulation model with all observations available (e.g. Lellouche et al., 2021; Forget et al., 2015). The numerical models have the advantage to offer a product that is physically consistent but can contain drifts and biases (Stammer, 2005). The data assimilation is a practical mean to reduce the spurious model drifts and biases but still, the model can diverge from observations and even drift in uncharted states in poorly sampled regions (Forget et al., 2015). Second, the statistical approach aims at finding the empirical relationship between the surface ocean and the interior. The simplest method is to use a multiple linear regression between SLA, SST and T-S profiles (Guinehut et al., 2012; Jeong et al., 2019). According to Guinehut et al. (2012) this method can only reconstruct 50 % to 30 % of the temperature and 20 % to 30 % of the salinity at depth. An improvement of the linear reconstruction method is to first reduce the T-S profiles, and to link up the reduced variables to the satellite data. Indeed it was found that only a few modes are needed to explain most of the variance/covariance of the temperature fields (Meinen and Watts, 2000) or of combined T-S profiles using the gravest empirical mode (GEM) projection (Sun and Watts, 2001). The GEM technique is a projection of hydrographic profiles onto a geostrophic stream function plane, which was used to estimate the four-dimensional structure of the Southern Ocean (Meijers et al., 2010). However it requires that each dynamic height be associated with just one T-S profile at each longitude, meaning that outside of the Antarctic Circumpolar Current or boundary currents, the approach is questionable. Buongiorno-Nardelli and Santoleri (2005) developed the multivariate Empirical Orthogonal Function Reconstruction (mEOF-r) based on a similar idea. It is a linear system that uses surface data to predict the three leading mode of the EOFs applied on profiles of temperature, salinity, and geopotential thickness. They later showed that mEOF-r is outperformed by an artificial neural network for the North Atlantic region (Buongiorno Nardelli, 2020).

Machine learning approaches are increasingly used to deal with the ever-growing stream of geospatial data (Reichstein et al., 2019; Sonnewald et al., 2021; Wang et al., 2019). More specifically, deep learning methods are characterized by artificial neural networks (NNs) involving usually more than two hidden layers. They exploit feature representations learned exclusively from data (Zhu et al., 2017). Multiple studies recently presented deep learning methods for reconstructing hydrographic profiles from satellites. Proof of concept papers established the important capabilities of feedforward or long short-term memory (LSTM) neural networks for hydrographic profiles predictions (e.g. Lu, 2019; Jiang et al., 2021; Contractor and Roughan, 2021; Buongiorno Nardelli, 2020; Su et al., 2021; Sammartino et al., 2020). NNs can also efficiently reconstruct Argo interpolated fields (Gou et al., 2020; Meng et al., 2021). A recent study focused on predicting the mixed layer depth (MLD) from satellites




using probabilistic machine learning (Foster et al., 2021). But to our knowledge these deep learning studies do not explore the vertical coherence of the predicted profiles, i.e. the presence of density inversions and the accuracy of the MLD prediction. The presence of density inversions makes an ocean product more difficult to use to initialize regional forecast models. Statically unstable profiles have to be removed when using the product to analyze ocean dynamics (e.g. New et al., 2021). The accuracy of the MLD prediction has also large implications for the pertinence of an ocean product. Indeed, the MLD and the strength of underlying stratification regulates the rate at which the ocean exchanges with the atmosphere, directly impacting our climate (Sallée et al., 2021). To understand and quantify the ongoing climate change, we need to document the variability of the vertical gradients of temperature, salinity and density in the water column. Physically consistent products, in the spirit of the MIMOC climatology (Schmidtko et al., 2013) but with higher resolution in time and space, are required to validate the models used for climate projections. In particular, western boundary currents such as the Gulf Stream play a major role in climate variability by carrying warm and salty near-surface waters northwards (Smeed et al., 2018) and directly warming the atmosphere (Minobe et al., 2008). In the present paper we focus on the Gulf Stream region for its challenging high variability and its large *in situ* sampling coverage.

Here we present a method to estimate the ocean stratification from surface data and the associated uncertainty using a prediction model fitted with *in situ* historical data. We train a NN to predict temperature and salinity (T-S) profiles down to 1000m and the MLD, in the Gulf Stream region, from satellite data covering 1993 to 2019. Our goal is to combine MLD, T and S predictions to produce T-S profiles that are physically consistent. The training is done on raw *in situ* profiles alone (not interpolated fields) and the prediction are generalised for daily gridded fields at ¼° horizontal resolution. Our framework also delivers a quantification of the uncertainty through a confidence interval of the model's prediction as well as the relative importance of each input variables. The proposed reconstruction method and resulting product are named OSnet for Ocean Stratification network.

The paper is organised as follows. Section 2 introduces the datasets used as inputs and outputs of OSnet, as well as the products used as benchmark to evaluate the performance of our reconstruction. Section 3 presents the method composed of the neural network and the MLD adjustment. Section 4 evaluates the accuracy of OSnet predictions, and presents property maps and sections. OSnet profiles are compared to a mooring, we also compare timeseries and an analysis of the relative importance of each input for each output. In section 5 we explore the potential of OSnet by estimating profiles from synthetic satellite data. Our conclusions are presented in Section 6.

## 2 Data

### 2.1 Temperature and salinity *in situ* profiles

We use the *in situ* temperature and salinity (T-S) vertical profiles from CMEMS quality controlled Coriolis Ocean Dataset for Reanalysis (CORA) database (Cabanes et al., 2013; Szekely et al., 2019). We keep only the profiles extending at least from 25 m to 1000 m for the period 1993 to 2019, totalising 67767 T-S profiles for the region 80°W to 30°W and 23°N to 50°N. All the profiles that do not reach 1000 m or start deeper than 25 m are discarded. The profiles are interpolated on an uneven





vertical grid with 51 levels, with the spacing increasing with depth (27 levels are within the first 100 m leading to a vertical resolution of 1 m in the upper levels and 450 m at 1000 m depth). The profiles with no data at the surface are extrapolated by repeating the shallowest observation point. There is little seasonal bias in the distribution of data with 5647 profiles by month

on average, a minimum of 5027 in February and a maximum of 6257 in October. The spatial distribution of profiles kept in the analysis is shown on figure 1a. It reveals a general lack of data in the center of the subtropical gyre compared to the Gulf Stream region west of 60°W. The temporal distribution (Fig 1b) reveals a significant increase of sampling after 2000 thanks to the Argo program (Wong et al., 2020). After 2012, the amount of T-S profiles stabilises at ∼400 profiles per months.

## 2.2  Input data

The input data (Table 1) are the surface satellite data and the bathymetry. The Mean Dynamic Topography (MDT) is the product of Centre National d'Etude Spatiale (CNES-CLS18) (Mulet et al., 2021) and the bathymetry is the ETOPO1 bedrock, distributed by NOAA (2009). The Sea Surface Temperature (SST) is from the European Space Agency (ESA) Climate Change Initiative (CCI) and Copernicus Climate Change Service (C3S), v2.3 and level 4 product. It provides gap-free maps of daily average SST at 20 cm depth and 0.05° x 0.05° horizontal grid resolution, using satellite data from the Along Track Scan-

ning Radiometer (ATSRs), Sea and Land Surface Temperature Radiometer (SLSTR) and the Advanced Very High Resolution Radiometer (AVHRR) series of sensors (Merchant et al., 2019). The Sea Level Anomaly (SLA) is the level 4 daily product from CNES-CLS, the 6.2_DUACS_DT2018. It has a 0.25° x 0.25° horizontal grid resolution. We also use geostrophic surface velocities derived from the SLA product (Table 1).

## 2.3  Additional datasets

We validate our results against other observational and synthetic datasets. The Sea Surface Salinity (SSS) CCI dataset is distributed at 0.25° x 0.25° horizontal grid resolution from 2010 to 2019 (Boutin et al., 2021). We do not use SSS as an input variable for several reasons. SSS satellite observations only cover the period 2010-2019 and its quality is questionable in high latitudes and cold water (Boutin et al., 2021). As our prediction only depends on the input variables, it is risky to rely on data containing systematic errors. Moreover we tested an architecture with the SSS as input and the results were not improved

significantly, only the surface salinity was slightly better. The relative importance algorithm also showed that SSS was not used significantly to make the prediction. It was of the same order of importance than the geostrophic currents (see section 3.4 on the explainability of the NN). However SSS is a useful product to compare with, and we discuss further this product compared to surface salinity in the result section 4.5 and Appendix B.

The global eddy-resolving reanalysis Glorys12 (Lellouche et al., 2021) is based on the physical model NEMO (Madec,

2015) and ocean observations assimilated by means of a reduced-order Kalman filter. It is provided at 1/12° horizontal grid resolution and daily mean. We also compare our results to the observation-based Armor3D weekly product (Guinehut et al., 2012). It is built in two steps, the first one is the prediction of synthetic T-S fields by multiple linear regression from SST and SLA. Then, in a second step, these T-S fields are combined to *in situ* T-S profiles with an optimal interpolation method. A section of OSnet is compared with the hydrographic section AT20 sampled along the 52.3°W meridian by the research vessel



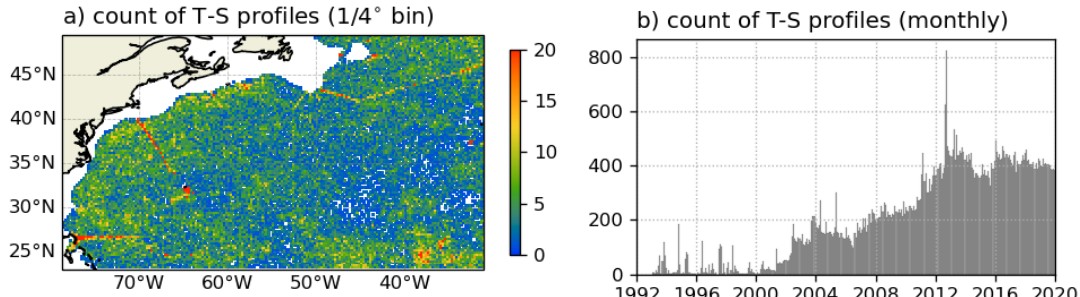

**Figure 1.** Count of temperature and salinity profiles extending from 25 to 1000 m for the region. The profiles are counted for 1/4° bins (a) and represented where it exceeds 0 with a truncated colorbar at 20 profiles, the maximum number of profiles for a bin is 185 profiles. The count in (b) is by months.

Atlantis from 2012-05-01 to 2012-05-11 (McCartney, 2012). Finally we use T-S data sampled at moorings of the Line W array, installed in April 2004 between Cape Cod and Bermuda. We use profiles from the third mooring located at 69.11°W, 38.51°N.

| Variable input | Temporal coverage | Distributor, version (citation) |
|---|---|---|
| Longitude | na | na |
| Latitude | na | na |
| Day of the year (cosine and sine) | na | na |
| Bathymetry | na | NOAA (ETOPO1 Bedrock) (NOAA, 2009) |
| Mean Dynamic Topography (MDT) | na | CNES-CLS18 (Mulet et al., 2021) |
| Sea Surface Temperature (SST) | 1981-ongoing | ESA CCI and C3S, v2.3, L4 (Good et al., 2019) |
| Sea Level Anomaly (SLA) | 1993-ongoing | CNES-CLS, 6.2_DUACS_DT2018, L4 |
| Zonal Absolute Geostrophic Velocities | - | - |
| Meridian Absolute Geostrophic Velocities | - | - |
| Zonal Geostrophic Velocities Anomalies | - | - |
| Meridian Geostrophic Velocities Anomalies | - | - |
| Additional datasets used | - | - |
| Sea Surface Salinity (SSS) | 2010-2019 | ESA CCI, v2.31 (Boutin et al., 2021) |
| Armor3D | 1993-ongoing | CMEMS (Guinehut et al., 2012) |
| Glorys12 | 1993-2019 | CMEMS (Lellouche et al., 2021) |

**Table 1.** List of input variables for the neural network and additional datasets to compare with. ESA CCI = European Space Agency Climate Change Initiative; C3S = Copernicus Climate Change Service ; NOAA = National Oceanic and Atmospheric Administration ; DUACS = Data Unification and Altimeter Combination System, CMEMS = Copernicus Marine Environment Monitoring Service.





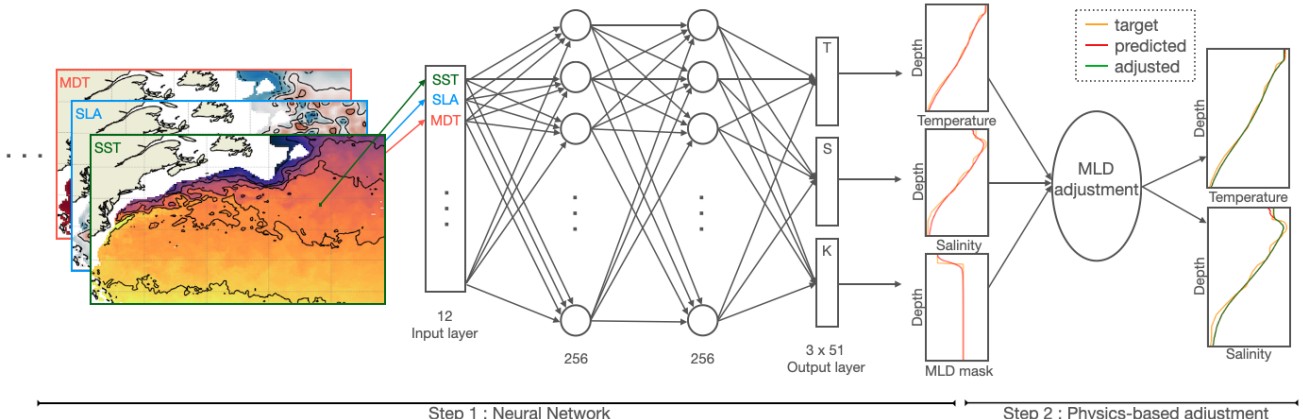

**Figure 2.** Schematic of OSnet formed of a Neural Network (NN) with two hidden layers and a mixed layer (MLD) depth adjustment. The NN uses 12 surface inputs (SST, SLA, MDT,...) that are listed in table 1, to predict profiles of temperature (T), salinity (S) and MLD mask (K). T-S profiles are then adjusted using the profile K for a better prediction of the MLD.

## 3 Method

The method is composed of two steps (Fig. 2). Firstly, a neural network predicts T-S profiles and the MLD from satellite data. Secondly, an adjustment of the MLD combines the predicted T, S and MLD to correct the vertical shape of the profiles
towards a physically consistent solution. The procedure is coded in Python with the help of several useful modules. The NN algorithm is coded with Tensorflow (Abadi et al., 2016) and the Keras application programming interface (Chollet and Others, 2015). It is explained with Shap (Kaur et al., 2020). Xarray (Hoyer and Hamman, 2017), Dask (Rocklin, 2015) and Numba (Lam et al., 2015) are used for fast computation and the management of large datasets. The color palettes for maps are imported from Cmocean (Thyng et al., 2016) and Colorcet (Kovesi, 2015). All the codes to build OSnet models are available
at https://github.com/euroargodev/OSnet and the models and prediction tools specific to the Gulf Stream region are available at https://github.com/euroargodev/OSnet-GulfStream.

### 3.1 The Multilayer Perceptron Neural Network

The neural network used is a multilayer perceptron (MLP), which is a class of feedforward artificial neural networks (Rosenblatt, 1961). The architecture is kept simple with only two layers of 256 neurones each. Dropout is used as a regularisation
method to reduce overfitting and improve generalisation (Srivastava et al., 2014). The activation functions are rectified linear activation function (ReLU) for the hidden layers, linear for T and S output, and a sigmoid for the MLD output. We tested more complex architectures (additional layers, convolutional layers, bottleneck architecture) but could not improve the accuracy of the results. A simple architecture is advantageous for its lower computation time.

The input consists of 12 values, listed in table 1: latitude, longitude, day of the year (cosine and sine), bathymetry, MDT,
SST, SLA, four geostrophic velocities U, V and both anomalies. The inputs are linearly interpolated at each *in situ* profile's



location. The output are prediction of three vectors of 51 depth levels (temperature, salinity and the MLD mask). The depth levels are the one presented in the data section, on which the CORA profiles are interpolated on.

The dataset is split (randomly with no replacement) as follows: 20% of the profiles are set aside for the test. In the remaining 80%, we use 80% as training data and 20% as validation data. The validation data is used to avoid overfitting by assessing the
performance of the trained model after each epoch (one epoch is seeing all the training data). Once the training of the NN is finished, we select the model with the best performance on the validation dataset. We then run this model on the test dataset (data not seen in training or validation) to confirm the good generalization of the model (i.e. training and test errors are similar, Fig. 3). Given a NN architecture with good generalization properties, we retrain a NN using all the 67767 profiles (Fig. 3, orange) as training data. The training of one model takes $\sim 20$ minutes on an 8 cores CPU with 32 Go of RAM.
To further improve the prediction performance and assess the associated uncertainty, we exploit a bootstrapping scheme (Breiman, 1996). More precisely, we bootstrap the training procedure 15 times using a different initialization and training dataset each time. Indeed because of the instability of the prediction method, the bootstrap can give substantial gains in accuracy. Overall, given 15 trained models, we compute the mean T, S and K profiles for each input data and the associated standard deviations (Fig. 3, grey). The latter deliver an estimation of a confidence interval. This bootstrap method reduces the estimation
bias. Finally, the T-S prediction is generalised on a daily ¼° horizontal grid. The spatial resolution of the input data (Table 1) is unified to ¼° in longitude and latitude by a nearest neighbour interpolation method. This produces a homogenised field of T and S with 51 depth levels from the surface to 1000 m for each day from the $1^{st}$ of January 1993 to the $31^{st}$ December 2019. It is freely available on Zenodo (Pauthenet et al., 2022).

## 3.2 Prediction of the mixed layer depth

We define the mixed-layer depth $H$ with a density deviation from the surface as proposed by de Boyer Montegut et al. (2004). It is the depth at which the potential density referenced to the surface, $\sigma_0$, exceeds by a threshold of $0.03$ kg m$^{-3}$ the density of the water at 10m, $\sigma_0(z = -H) = \sigma_0(z = -10m) + 0.03$ kg m$^{-3}$. This definition is chosen for its simplicity of application, but tends to overestimate deep winter MLDs compared to the more sophisticated hybrid algorithm of Holte et al. (2017). These regions of deep winter MLD are rarely observed in our dataset (1145 profiles with MLD deeper than 300m, i.e. $1.7\%$ of the
dataset), comforting our choice of using the simple density threshold. For the NN, the mixed-layer is represented in the form of a unitless profile $K$ of size $Z = 51$, that is filled with zeros between the surface and the mixed layer depth $H$, and with ones from $H$ to $-1000$ m :

$$K(z) = \begin{cases} 0 & \text{if} \quad z > H \\ 1 & \text{if} \quad z \leq H \end{cases} \tag{1}$$

This form is preferred because it allows the NN to give an estimation of the gradients around the MLD instead of a single depth value (Fig. 4d). The resulting mask $K$ is also convenient for the MLD adjustment performed on the predicted profiles.





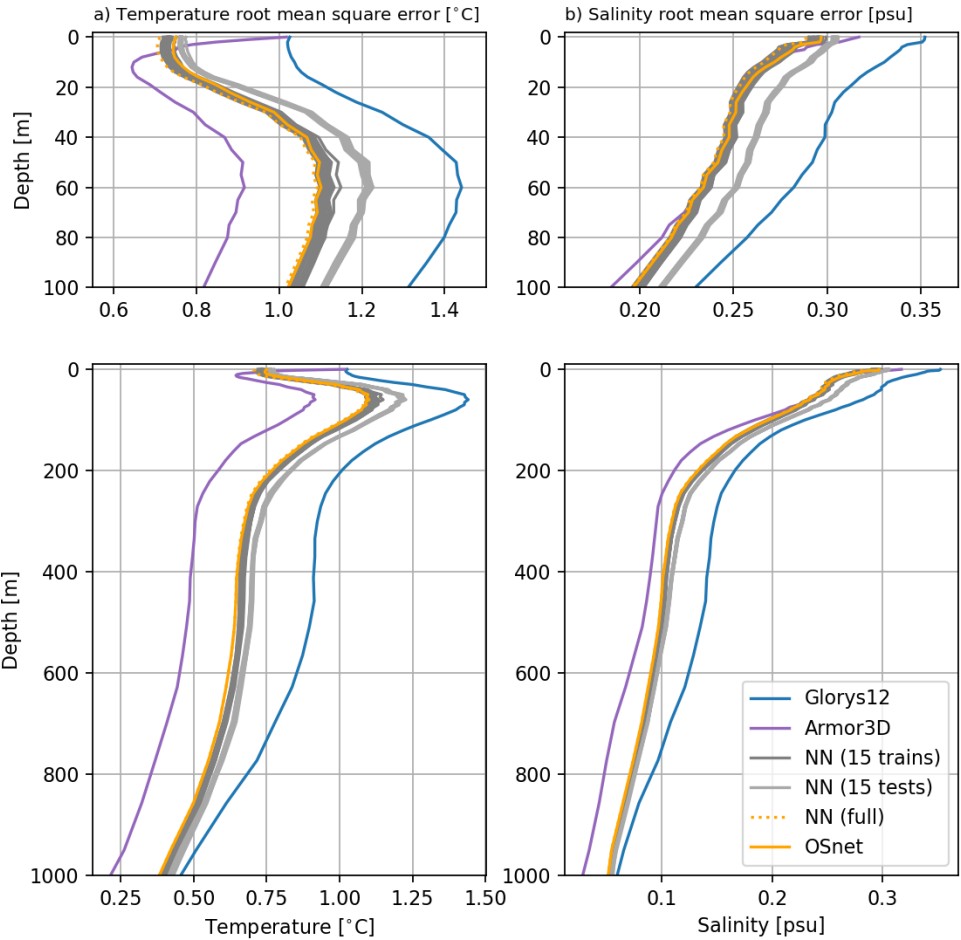

**Figure 3.** Root Mean Square Error between temperature (a) and salinity (b) observed (CORA) and predicted profiles (Glorys12, Armor3D, NN and OSnet). The upper panels are a zoom of the first 100 m of the full depth lower panels. The Glorys12 (blue) and Armor3D (purple) profiles are collocated with the CORA profiles and the error is calculated between these subsamples. The NN profiles are only predicted with the NN, without adjustment of MLD, for 15 train datasets (dark grey) and 15 test datasets (light grey). The NN full (orange dotted) correspond to the predictions using the full dataset (test + train) and averaged for 15 run (bootstrap). Finally the OSnet profiles (orange) are predicted with a NN bootstraped 15 times and the MLD adjustment is performed, which slightly increases the error at the surface.

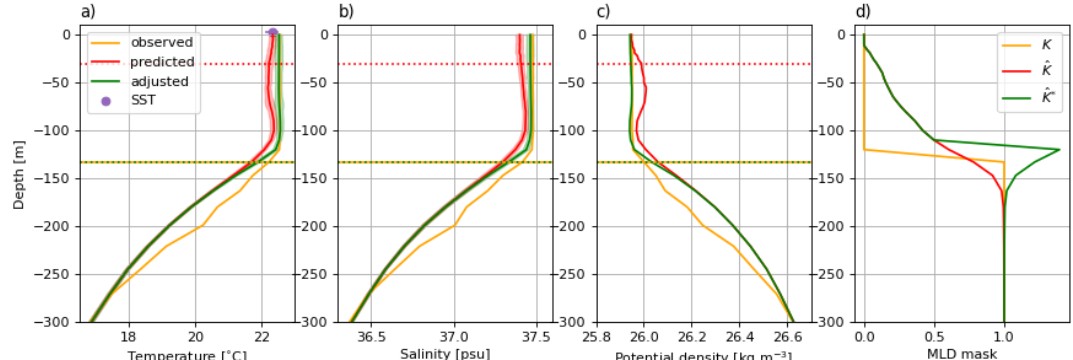

**Figure 4.** Example of profile (orange) sampled the 2012-03-03 at $35.55°$W and $23.53°$N, truncated at 300 m deep. We display temperature (a), salinity (b), potential density (c) and the profile $K$ that is a mask of MLD (see equation (1)). The T-S profile predicted by the NN is in red and the adjusted using the $\hat{K}^*$ profile (OSnet) in green (see section 3.3). The green and red bands are the confidence interval for each profiles. Mixed Layer Depths (MLDs) for observed, predicted and adjusted are shown with dotted horizontal lines. SST is added with a purple dot and a horizontal bar for its mapping uncertainty.

## 3.3  Adjustment of the mixed layer

As explained previously, the NN predicts three vertical profiles; $\hat{T}$, $\hat{S}$ and $\hat{K}$, the latest being an estimation of a profile $K$ filled with zeros in the mixed layer and ones below the MLD. The prediction of profile $K$ can be seen as a classification problem, and outputs at each depth the probability to be either above or under the MLD. Hence the predicted MLD is located at the depth where $\hat{K} = 0.5$ if the probability distribution of MLD is symmetrical. We will see later that it is not exactly $0.5$ for our region and we introduce a threshold variable $\lambda$ for which $K = \lambda$ corresponds to the MLD. $\hat{K}$ is a sigmoid-like profile (Fig. 4d) and its vertical gradients are proportional of the T-S gradients around the MLD. Indeed, the summer $\hat{K}$ profiles have sharp vertical gradients compared to the winter $\hat{K}$ profiles (not shown) which is coherent with the seasonal cycle of the transition layer thickness (Johnston and Rudnick, 2009).

The MLD predicted by the $\hat{K}$ profile has a better accuracy (MLD RMSE of 40 m) than the MLD computed from the T-S profiles directly (MLD RMSE of 50 m). The latter is systematically too shallow due to unrealistic T-S excursions on the vertical in the mixed layer, causing the density threshold to be reached too close to the surface. These sharp variations of T and S in the mixed layer also creates density inversions.

A second bias is identified on the predicted profiles. The gradients for the layer under the MLD are systematically underestimated compared to the observed profiles. The mean and standard deviation of gradients of $\sigma_0$ over a 200 m thick layer under the MLD is of $1.33 \pm 0.9$ kg m$^{-3}$ for the observed profiles and $1.24 \pm 0.8$ kg m$^{-3}$ for the predicted ones. The presence of strong gradients under the MLD has been documented (Johnston and Rudnick, 2009) and the profile $\hat{K}$ seems to contain this information.



After comparing different methods we thus choose to apply a MLD adjustment on the predicted profiles, in the same spirit as convective adjustment schemes used in numerical hydrostatic models (Madec, 2015). We want to weight the vertical gradients

of $\hat{T}$ and $\hat{S}$ by $\hat{K}$, in order to reduce the gradients of $\hat{T}$ and $\hat{S}$ in the mixed layer, increase the gradients of $\hat{T}$ and $\hat{S}$ just below the MLD ($\hat{K} = \lambda$), while keeping the deeper gradients unchanged. So we modify the $\hat{K}$ into a new mask $\hat{K}^*$ as follows :

$$\hat{K}_z^* = \left\{ \begin{array}{lll} \hat{K}_z & \text{if} & \hat{K}_z < \lambda \\ 2 - \hat{K}_z & \text{if} & \lambda > \hat{K}_z > 1 \\ \hat{K}_z & \text{if} & \hat{K}_z = 1 \end{array} \right. \tag{2}$$

Then we recompute iteratively the T-S profiles with the gradients adjusted with $\hat{K}^*$, starting by the bottom value (z = 1000m) where $\hat{K}^* = 1$ because the deepest MLD is never reaching 1000m (the maximum observed in the CORA dataset is $H = 628$m for our region). On a predicted temperature profile $\hat{T}$ (the same is applied to salinity), the adjusted temperature profile $\hat{T}^*$ is

computed as follows :

$$\frac{\Delta \hat{T}^*}{\Delta z} = \frac{\Delta \hat{T}}{\Delta z} \hat{K}^*, \tag{3}$$

and we retrieve the temperature profiles iteratively along depth, by starting from the bottom $z + 1 = 1000$m, where the $\hat{K}_{z+1}^* = 1$ and $\hat{T}_{z+1}^* = \hat{T}_{z+1}$ :

$$\hat{T}_z^* = \hat{K}_z^* \left( \hat{T}_z - \hat{T}_{z+1} \right) + \hat{T}_{z+1}^* \tag{4}$$

The adjustment of gradients above and under the MLD, by using the modified profile $\hat{K}^*$ is justified by looking at figure 5. The direct prediction of temperature at 0m (Fig. 5a, blue) is more accurate compared to SST than *in situ* observations, because

OSnet learns from SST. The adjusted profiles with the predicted $K$ profile contain a warm bias of the surface temperature (Fig. 5a, red). By adding the constraint on the gradients under the MLD ($\hat{K}^*$), the error between SST and surface temperature is centred, close to the observations. The salinity difference relatively to the SSS is too large for the adjustment to cause a significant issue (Fig. 5b). Still the adjusted salinity profiles with $K$ predicted creates a fresh bias and the use of $\hat{K}^*$ corrects that. For the calibration of parameter $\lambda$ in (2), we use a cross-validation procedure according to the estimation bias between $\hat{T}$

at sea surface and the SST value (green in Fig. 5). This leads to $\lambda = 0.57$. We expect this value to be specific to our region and of the considered NN parameterization. It would likely require a new calibration for other case-study regions.

A good example of profile prediction and adjustment is presented on figure 4. In this case the adjustment corrects perfectly the MLD estimation, from 30 m predicted (red) to 133 m adjusted (green). It reduces the T-S gradients above the MLD estimated from the K profile (K = 0.5), and increases the T-S gradients just under this MLD. The decrease of T-S gradients in

the mixed layer also removed the density inversion for this case (Figure 4c). This profile is responding particularly well to the adjustment. The adjustment proposed here reduces the variance of $\hat{T}$ and $\hat{S}$ above the MLD, reduces the number of density inversion, improves the predictions of MLD (see table 2) and increase the gradients in the transition layer under the MLD. For the adjusted profiles, the mean and standard deviation of gradients of $\sigma_0$ over a 200 m thick layer under the MLD is of $1.38 \pm 0.9$ kg m$^{-3}$ which is closer to the observed values of $-1.33 \pm 0.9$ than the predicted values of $1.24 \pm 0.8$ kg m$^{-3}$. The

summer high gradients are especially well retrieved (not shown).





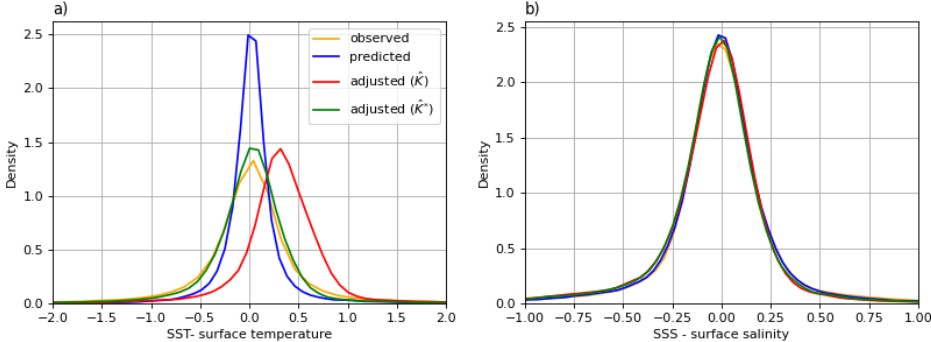

**Figure 5.** Density distribution of the difference between the *in situ* surface temperature and salinity and the remote sensing SST and SSS. The predicted profiles (blue) corresponds to the profiles produced by the NN alone. The adjusted distribution with $\hat{K}$ are in red (also red in figure 4d). The adjusted distribution with $\hat{K}^*$ is in green and corresponds to the OSnet product (i.e. NN + adjustment), also shown in green in figure 4d.

An alternative idea to solve the density inversion issue is to constrain the NN to predict profiles that are hydrostatically stable. The physical relationship can be implemented in the NN to enforce consistency on the predictions (Karpatne et al., 2017). This can be done by modifying the loss of the NN and penalising the predictions with density inversions (Appendix A). This solution is elegant and allows to predict directly profiles without density inversions. We provide this alternative approach

here for the record of a negative result with regard to the design of such NN. Indeed, the profiles predicted with this custom loss still have poor MLD estimation compared to the $\hat{K}$ profiles, and need *a posteriori* adjustment. The modified loss (Appendix A) is not needed in this case because the MLD adjustment presented above happen to remove sufficiently the density inversions.

### 3.4   Explainability of the neural network

Explaining the predictive skills of the neural network is key to interpreting the prediction and strengthening trust in the model.

It is also a useful tool in the development phase. Here, it gives us insights into the relationship between surface data and *in situ* profiles. In this section, we use a game-theoretic approach to retrospectively estimate the relative importance of each input for each output. The algorithm, called SHapley Additive exPlanations (SHAP) (Lundberg and Lee, 2017) is a unified framework combining states of the art methods to explain deep neural networks. It is based on a method called Deep Learning Important FeaTures (DeepLIFT) (Shrikumar et al., 2017) and Shapley values. DeepLIFT is a method for computing the effect of changing

the original input to a reference value (uninformative background value for the input). The change in the output is representative of the importance of the input to predict the output. Shapley regression values (Shapley et al., 1953) are representing the impact of an input on the output by removing it from the input set and retraining this model with the subset of inputs. This being computationally expensive, it is possible to obtain an approximation of the effect of removing a variable from the model by integrating over samples from the training dataset using the Shapley sampling values (Štrumbelj and Kononenko, 2014).

It produces an "importance" value for each particular prediction. The importance value is positive or negative, indicating the




| | CORA | Armor3D | Glorys12 | NN | OSnet : NN + MLD adjustment |
|---|---|---|---|---|---|
| ratio of profiles with $\sigma_0$ vertical inversion (%) | 1.37 % | 53.72% | 0.01% | 17.3 % | 0.32% |
| mean size of $\sigma_0$ vertical inversion (kg m$^{-3}$) | $2.30 \times 10^{-3}$ | $8.85 \times 10^{-2}$ | $1.15 \times 10^{-4}$ | $8.19 \times 10^{-3}$ | $8.39 \times 10^{-4}$ |
| rmse of MLD | - | 39.6* | 39.3 | 50.0 | 40.0 |
| rmse of T | - | 0.488 | 0.878 | 0.658 | 0.661 |
| rmse of S | - | 0.093 | 0.141 | 0.110 | 0.110 |
| rmse of $\sigma_0$ | - | 0.073 | 0.104 | 0.082 | 0.083 |

**Table 2.** Metrics of accuracy for predictions of Armor3D, GLorys12, a Neural Network (NN) and and OSnet, compared to the *in-situ* CORA profiles. The $\sigma_0$ inversions larger than 0.01 kg m$^{-3}$ are counted. The Armor3D and Glorys12 statistics are computed on the subsampled products at CORA's profile locations. The mean and the standard deviation is provided. (*)All the MLDs are computed with the density criterion of 0.03, except for Armor3D for which a different criterion is used to bypass their density inversion issues.

direction in which the input influences the output, relative to the averaged output. The SHAP algorithm being computationally expensive it was not possible to run it over the full dataset. After some tests, we found that 300 random samples were representative enough to obtain stable results for the average feature of importance across the entire dataset.

## 4 Results

### 4.1 Accuracy of the predictions compared to observations

Table 2 presents different metric to evaluate OSnet, Armor3D and Glorys12 relatively to CORA. Each dataset is predicted or subsampled at the location of CORA's profiles in longitude, latitude, depth and time. The two first lines indicate the number and size of vertical density inversions, indicating that the MLD adjustment of OSnet suppresses almost all density inversions, from 16.41% to 0.29% of profiles. Meanwhile, Armor3D has about 50% of profiles with density inversions and Glorys12 almost none (0.01%). Regarding the amplitude of the density inversions, the MLD adjustment suppresses well large inversions and decreases by one order of magnitude the mean amplitude of inversions, from $10^{-3}$ to $10^{-4}$ kg m$^{-3}$.

The RMSE of the MLD (table 2) indicates that the MLD adjustment improves the MLD RMSE of OSNet from 50 m to 40 m, which is of the same order of magnitude than Glorys12 (38.6 m) and Armor3D (39.4 m). Note that the MLD of Armor3D is computed with a different criterion to bypass the density inversions. They use the minimum of temperature and density threshold equivalent to a $0.2°C$ decrease from the surface. The MLD of Armor3D computed with the density criterion of $0.03$kg m$^{-3}$ yields a RMSE of 62.6 m. Finally the global errors of temperature, salinity and density indicate that Armor3D profiles are the closest to the observed profiles. OSnet has a smaller T-S error than Glorys12 and the MLD adjustment increases the temperature RMSE from $0.658°C$ to $0.661°C$. Overall, the RMSE of OSnet predictions is of the same order of magnitude compared to other products, and it does not contain significant density inversions.





## 4.2 Temperature and salinity maps

Let us examine a daily map of temperature and salinity at 1/4° resolution (Fig. 6). We chose a date in the pre-ARGO era to show the great generalisation of OSnet product. The maps reveal coherent horizontal structures. At the surface, the warm Gulf Stream is detaching from Cape Hatteras and meandering further East, transforming into the North Atlantic Current (Fig. 6a,b). The surface confidence intervals are maximum for the cold and fresh waters near the edge of the continental slope and inside the cold and fresh core eddies and meanders (Fig. 6c,d). On average the confidence intervals are highlighting the cold waters north of the Gulf Stream (Fig. 7a,b) which is consistent with the error of prediction maps presented in Buongiorno Nardelli (2020). This could be due to the lack of profiles containing these cold waters in our dataset. At depth (1000 m in Fig. 6e,f), the signature of large eddies are visible, associated with a maximum of confidence interval again (Fig. 6g,h). The salty and warm Mediterranean Overflow Waters are present in the south east of the region. The average confidence intervals at 1000 m are highlighting the Gulf Stream and its meanders, rather than the water north of the Gulf Stream like at the surface (Fig. 7c,d). It corresponds to the areas with the largest variability (Gaillard et al., 2016; Forget and Wunsch, 2007). Note the different colorbar between the surface and 1000 m, the confidence interval large values at depth are twice smaller for temperature and five times smaller for salinity.

## 4.3 Mixed Layer Depth maps

To illustrate the quality of the predicted MLD of OSnet, we show MLD maps for a given day (2018-01-05) in figure 8. We pick a winter day to display the deep MLD areas. The direct prediction of the NN (Fig. 8a) has shallow patches in a few places that are due to the density inversions. The density threshold is met too shallow due to these artefacts in the water column (see profile example on the figure 4). The MLD adjustment is correcting well these shallow patches and the MLD fields of OSnet look more consistent, and more similar to Glorys12. The OSnet MLD does not exhibit any very deep patch (MLD > 300m). These deep MLD event are rarely observed in CORA (1.7% of the profiles have a MLD deeper than 300m) but are often present in the MLD fields from Glorys12 (Fig. 8). They occur in warm meanders and eddies of the Gulf Stream. The MLD field of Armor3D (Figure 8d) is for the week that contains the day 2018-01-05 and has several patches of either shallow or deep MLD (i.e. 28°N, 52°W or 48°N, 32°W), that look very sharp, compared to OSnet and Glorys12. These patches might be regions around observed profiles for the given week and the optimal interpolation of Armor3D is overfitting the profile, at the expense of the general property field coherence. The Armor3D MLD is computed with a different criterion to bypass their density inversion issues. Still, some patches have no values where the criterion could not be matched (Figure 8d).

Monthly MLD averages are presented for March and August in figure 9. The average in $1° \times 1°$ boxes for the *in situ* profiles (Fig. 9a,d) are compared to the OSnet (b,e) and Glorys12 (c,f). The three estimations are in good agreement with each other and with other climatologies (Holte et al., 2017; Sallée et al., 2021). The main structures are well respected with a large winter patch of deeper MLD extending between the Gulf Stream and the subtropical gyre. In winter, vigorous air-sea fluxes extract heat from the ocean and erode the superficial stratification. This process activates convective mixing and deepens the mixed layer, ventilating and creating the Eighteen Degree Mode Water (Speer and Forget, 2013; Maze and Marshall, 2011).





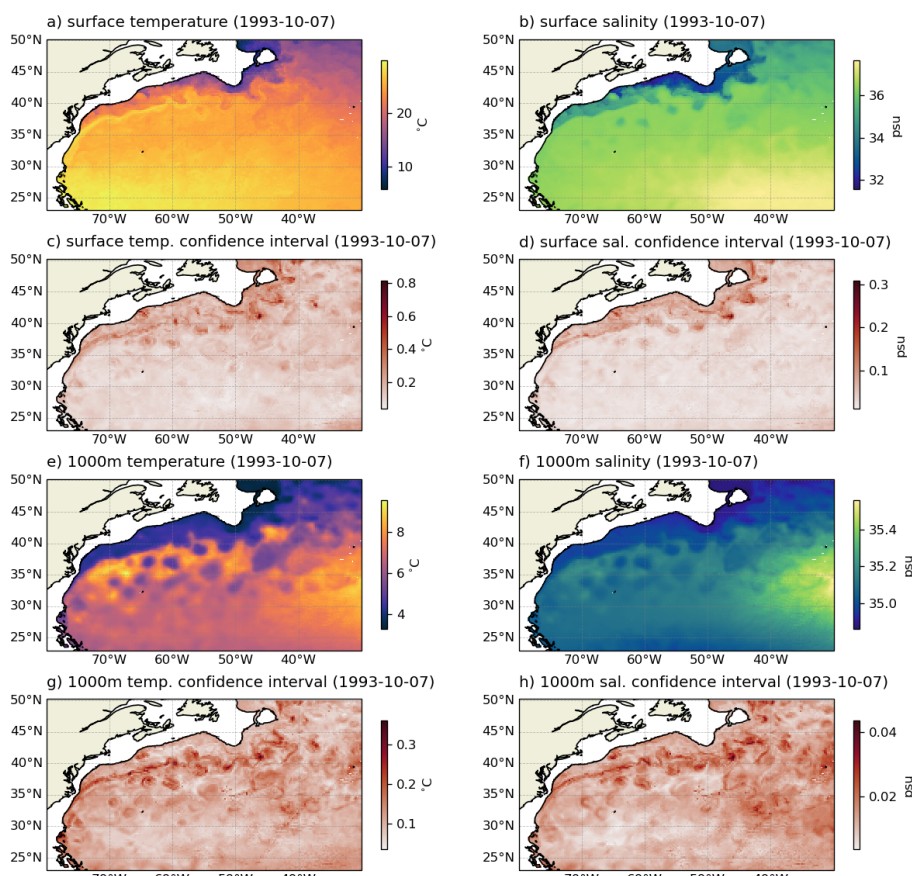

**Figure 6.** OSnet temperature and salinity maps for the date 1993-10-07, at the surface (a,b) and at 1000 m (e,f). Their respective confidence interval are displayed too (c,d,g,h).



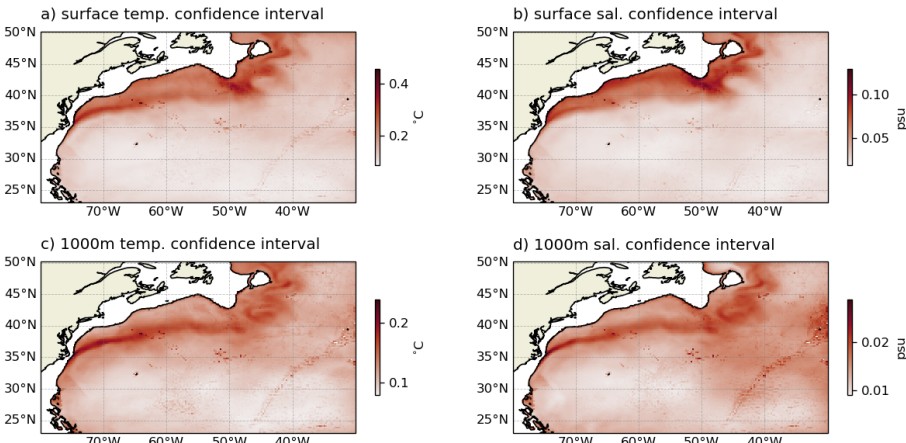

**Figure 7.** Time average maps of the confidence intervals for surface and 1000 m temperature and salinity.

In summer the near surface water warms and caps the mode water layer. The summer MLD is shallower everywhere with a slightly deeper signature in the core of the Gulf Stream, as it separates from the coastline at Cape Hatteras. A deeper summer
MLD is also found south of ∼ 30°N, along the equatorward edge of the subtropical gyre (Fig. 9d,e,f), a feature also observed in the climatology of de Boyer Montegut et al. (2004). This tropical summer MLD deeper than 30 m is marked by the trade winds (Stramma and Schott, 1999). The large permanent anticyclonic "Mann eddy" (Mann, 1967; Rossby, 1996) is clearly visible as a deep mixed layer patch at 43°N, 42°W (Fig. 9). A region of deeper MLD is also visible along the North Atlantic Current, deeper in OSnet than in Glorys12.

## 4.4 Importance of each input for the reconstruction

In figures 10 and 11 we present the absolute value of the relative importance of each input, on each output, averaged over depth, over the 300 test profiles, and over the 15 bootstrapped models. The error bars correspond to the standard deviation of the 15 models. To be comparable, the importance values of the inputs are normalized so that the sum per output is equal to one. The figures 10 and 11 give a general overview of what the NN uses for the predictions. These importance values can
also be displayed for a specific profile or by depth levels, seasons, or geographical regions, providing insights to elucidate the behaviour of the NN. The main result here is that SST is the main driver for estimating T, S and MLD profiles (Fig. 10). As expected it is especially important for predicting surface temperature. MDT is the second most important variable and it is the most important deeper than ∼ 200 m (Fig 11). The latitude, longitude, SLA and the day of the year equally concur to explain the predictions. The rest of the input variables, i.e. bathymetry and the surface geostrophic currents derived from SLA are
smaller contributions to the predictions. Even if they have small contributions on average, they can be important for a given





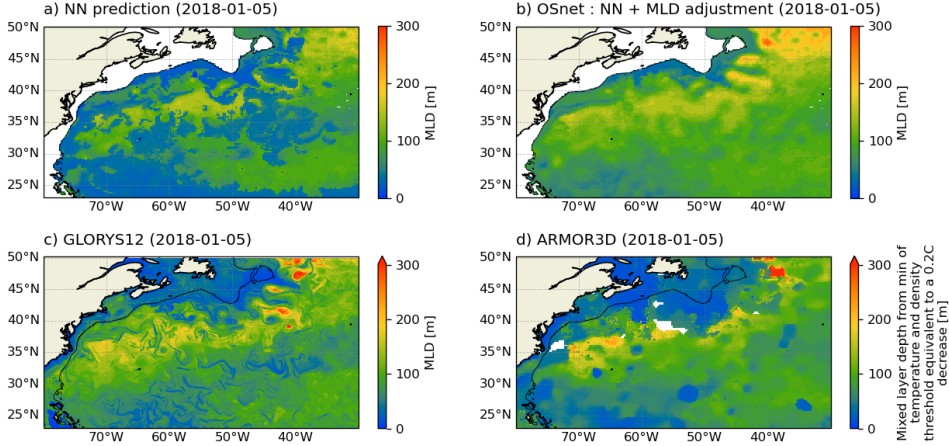

**Figure 8.** Mixed layer depth (MLD) maps for 2018-01-05. a) shows the result of the NN alone (step 1 from the schematic figure 2) and b) is the final OSnet product. Panels c) and d) are the MLD of the TS profiles of GLORYS12 and ARMOR3D. ARMOR3D is a weekly product so the profiles here are for the first week of January 2018. All the MLDs are computed with the density criterion of 0.03, except for ARMOR3D for which a different criterion is provided to bypass their density inversion issues. The shelf break is traced in black with the bathymetry contour of 1000m. The maximum of the colorbar is set by the maximum of OSnet.

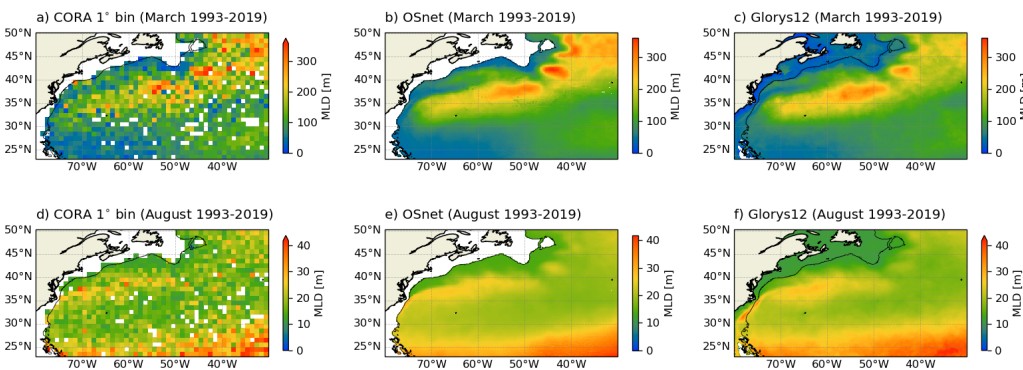

**Figure 9.** Maps of the monthly mean of the mixed layer depth (MLD) defined with a density threshold of 0.03 kg m$^{-3}$ for (a, d) CORA T-S profiles averaged by bins of 1°, (b,e) OSnet and (c,f) Glorys12. The months of March and August are representative of the seasonal range of the MLD for the region. Note the different colorbar range for each month. The maximum of the colorbar is set by the maximum of OSnet. The shelf break is traced in black with the bathymetry contour of 1000m.





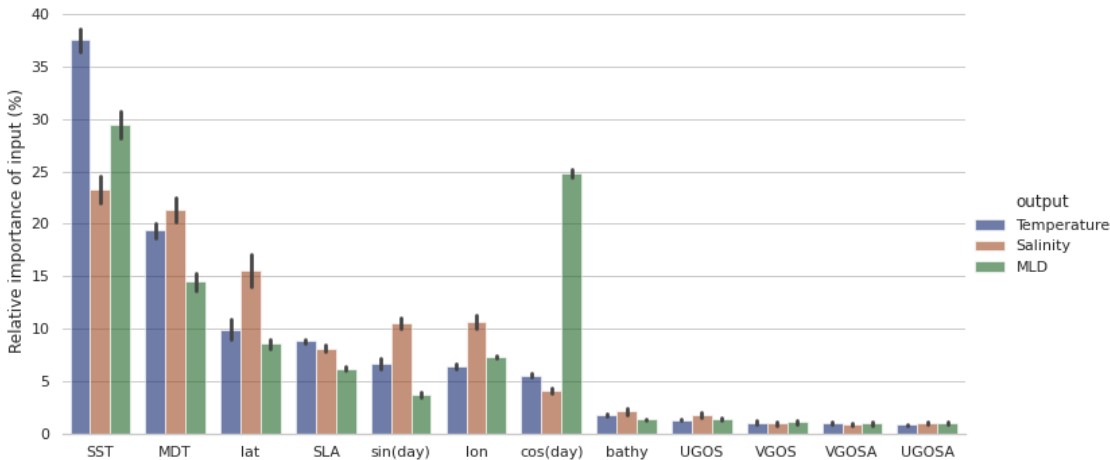

**Figure 10.** Relative importance of each input for each output, averaged by depth. The inputs (x-axis) are sorted by importance for the temperature, to have the largest importance on the left of the plot. The cosine of the day of the year is more important than the sine for the MLD prediction because the cosine is in phase with the seasonal cycle of the MLD.

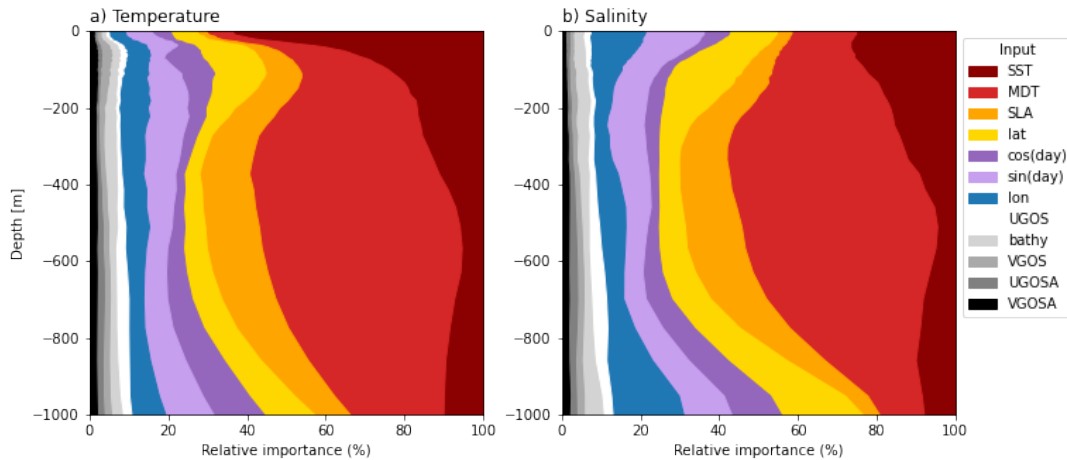

**Figure 11.** Relative importance of each input for each output, averaged by depth. The areas are sorted by variance to have the input with the largest difference of impact by depth to the right of each panels.

profile. The cosine of the day of the year is very important for the prediction of the MLD (Fig. 10), probably because it is in phase with the MLD seasonal cycle, while temperature and salinity cycles are in phase with the sine of the days (not shown). Still, it means that the day of the year alone drives $\sim 29\%$ of the MLD predictions, which is equivalent to the importance of SST on the MLD predictions ($\sim 29\%$ too).





### 4.5 Time series of surface properties


To assess the accuracy of OSnet through time, we analyse the time series of the spatially averaged surface temperature and compare it with the observed SST time series, as well as the Glorys12 and Armor3D products. We average the data over the region after removing the values over the shelf (bathymetry>1000m). The long term trends are obtained by applying a seasonal-trend decomposition based on loess (STL) (Cleveland et al., 1990). STL is a filtering procedure that extract three components:

(i) the variations in the data at the seasonal frequency (Fig. 13), (ii) the low frequency variation together with nonstationary and long-term changes (Fig. 12) and (iii) a remaining high frequency component.

OSnet follows best the long-term trend of SST with a linear trend of $0.197°$C/decade, close to the SST trend of $0.190°$C/decade (Fig. 12a). For comparison the surface averaged temperature of Armor3D is warmer in the pre Argo years, probably due to their background global average that is mostly composed of Argo floats. This is a validation that OSnet generalises well and is

not biased towards the recent years of *in situ* observations. Regarding the surface salinity long term trends, no significant trend is captured over the 1993-2019 period (Fig. 12b). There is no clear agreement between the different datasets, except in the last period 2010 to 2019, where all averages increase like the SSS signal. Armor3d mean surface salinity drops significantly during the last two years 2018 and 2019, out of the SSS error margin. Note that OSnet does not include the areas over the continental shelf and does not predict deeper than 1000 m. A large part of the climatic signal takes place in the coastal regions (Ezer et al.,

2013; Davis et al., 2017) and an improvement of OSnet would be to deal with profiles of different length in order to include these regions.

The mean seasonal variation of surface temperature, salinity and MLD is presented in figure 13. We compute it by averaging the signal by day of the year (week of the year for Armor3D). The surface temperature seasonal signal is well reproduced for each dataset, as expected considering that SST is included in the input of OSnet and used to produce both Armor3D and

Glorys12 (Fig. 13a). A close observation of the curves in figure 13a shows that the seasonal cycle of OSnet surface temperature is too cold between May and September (Fig. 13c). This is due to the MLD adjustment because the direct prediction of surface temperature with the NN gives a very precise seasonal cycle of SST (red line in fig. 13c)). Armor3D seasonal temperature is warmer from August to April, which could be a bias caused by the undersampling of the pre-Argo, colder years (Fig. 13c). This bias is also present on the temperature timeserie (Fig. 12a).

The surface salinity seasonal signal is noisier, in part because it presents inter-annual variations that are of the same order of magnitude than the seasonal variations (Fig. 13b). We only compare the 2010-2019 period because it is the only period available for the SSS product. OSnet surface salinity is closest to the SSS compared to the prediction of Armor3D and Glorys12, that are both fresher than the observed SSS. We also observe a delay in the SSS seasonal variation. It is fresher by almost 0.1 psu from January to March and we discuss that in Appendix B and figure A1.

Finally the seasonal cycle of MLD (Fig. 13d) in the region is asymmetrical, with slow deepening from summer to winter, and fast shoaling during the spring. This asymmetry is expected as it is fast to shoal the MLD with warming from the sun, rainfall, river outflow, or currents that slide lighter water over denser. In contrast, the deepening of mixed layer is slower and requires loss of heat or freshwater to the atmosphere or strong wind mixing to overcome the ocean's stratification (Johnson and





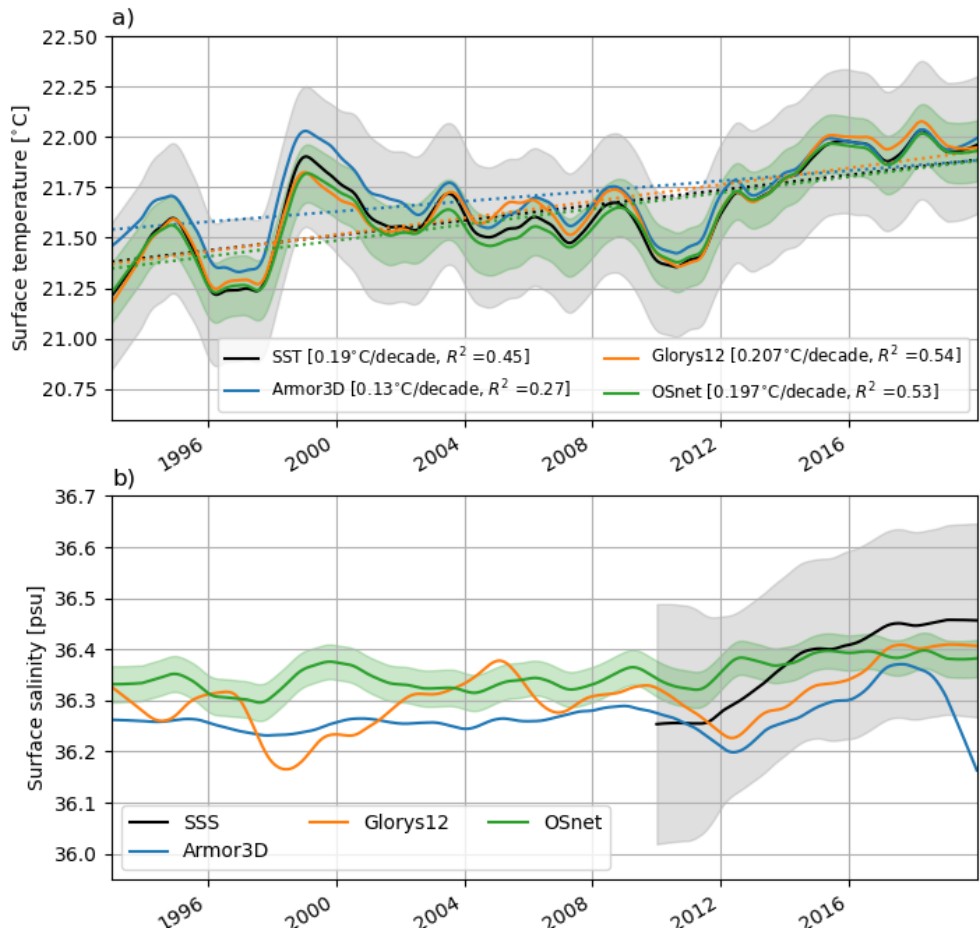

**Figure 12.** Nonseasonal low frequency timeserie of surface temperature (a) and salinity (b) averaged over the region excluding the shelf shallower than 1000 m. It is extracted with a seasonal-trend decomposition. The shaded grey areas are the SST mapping error in a), and the SSS random error b). The green shaded areas are the OSnet confidence intervals.

Lyman, 2022). OSnet MLD compares well with the MLD computed on Glorys12. We do not present the MLD of Armor3D here because it is computed with a different criterion. The winter MLD variance is larger in Glorys than OSnet, which is also observed on daily maps of MLD. Events of deep MLDs are not represented in OSnet (Fig. 8).




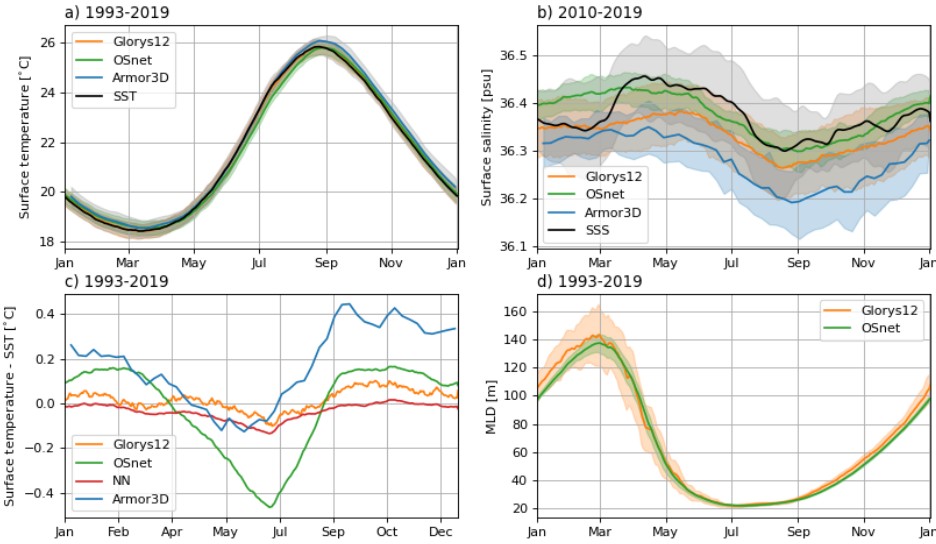

**Figure 13.** Seasonal variation of the surface temperature (a), surface salinity (b) and MLD (d) for OSnet, Armor3D and Glorys12, compared to remote sensing SST and SSS (black). The difference between SST and the surface temperature is displayed in c), with the direct prediction of the neural network (NN) hence without MLD adjustment in red. It is averaged over the period 1993-2019 excluding the shelf shallower than 1000 m, except for the surface salinity because SSS only ranges 2010-2019. The bands or errors are the standard deviation over the time period.

## 5 Discussion

In the results section we have seen that the OSnet predictions are coherent globally. We now want to assess if OSnet can be used to help interpret local oceanographic measurements or for process studies. To do so OSnet is compared with observations
(remote and *in situ*) and with the two other products Armor3D and Glorys12. In this section we present these comparisons and discuss the quality of our predictions.

### 5.1 Comparison with observed data

One convenient feature of OSnet is the possibility to estimate T and S profiles at any location, given that the surface data is provided. Here we predict T-S at the location of a mooring of the line W (Fig. 14) and along the hydrographic section AT20
(Fig. 15). Temperature and salinity at 1000m is plotted in comparison with the mooring data for a period of three years (2004-05-10 to 2007-03-11). OSnet corresponds well with observations, but with a slight warm and fresh shift in the first year (Fig. 14). A warm core eddy crosses the location of the mooring in 2006 (see SLA map in fig. 14a) and its warm and salty deep signature is well captured by OSnet. Smaller warm and salty spikes appear in October 2004, July 2005 and November 2005





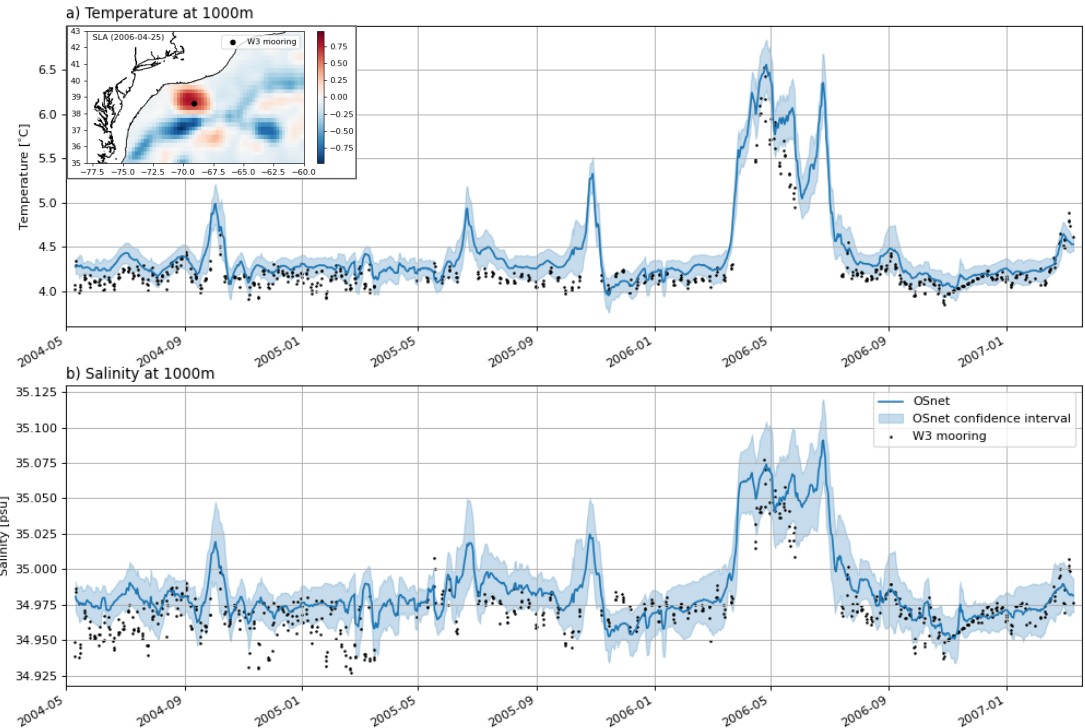

**Figure 14.** OSnet prediction of temperature and salinity at 1000m (blue) at the location of a mooring of the line W3 (black, $69.11°$W, $38.51°$N). The map of Sea Level Anomaly in the temperature panel is for the $25^{th}$ of April 2006 when a warm core eddy went through the mooring's location. This mooring data is not included in the learning dataset of OSnet.

but are not visible on the mooring data. They correspond to warm meanders of the Gulf Stream revealed by the SST and SLA

at these three periods (not shown), causing the T-S deep changes in OSnet.

We compare the OSnet T-S structure along the hydrographic section AT20 sampled in Mai 2012 by the research vessel Atlantis (Fig .15). The OSnet prediction is done at the exact location of the CTD profiles, interpolated linearly on the maps of input data (Table 1). The comparison is also done for Glorys12, but by colocating the profiles with a nearest neighbour method. Both Glorys12 and OSnet predictions are coherent but we acknowledge two specific differences. First, Glorys12 displays a

deep patch of MLD around $41°$N, north of the Gulf Stream, that is not observed by the CTDs, nor predicted by OSnet (Fig. 15d). This deep MLD could be due to the nearest neighbour selection of profile that is not exact in the case of Glorys12, or it could be an artefact of their model. Indeed deep patches of MLD are also visible on daily MLD maps of Glorys12 but are absent of OSnet daily MLD maps (Fig. 8). Second, the salinity reconstruction of Glorys12 differs more from the observed data north of the Gulf Stream, compared to OSnet (Fig. 15g). Salinity is more difficult to reconstruct than temperature, especially

in this highly variable region. OSnet performs well along this section, in comparison to the reanalysis Glorys12.




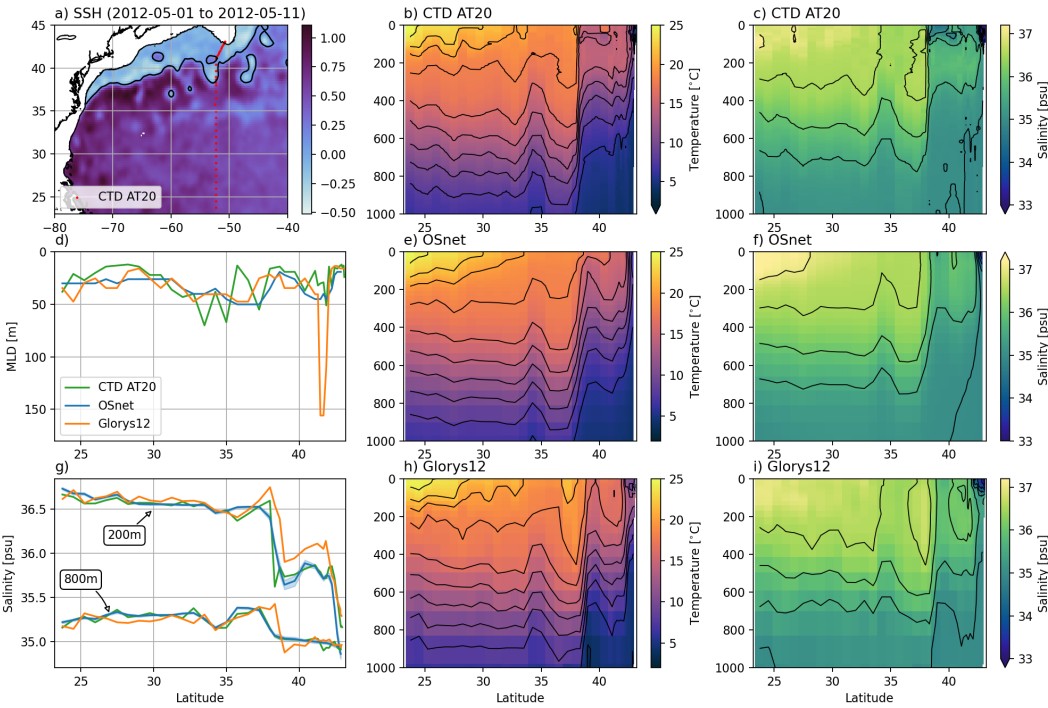

**Figure 15.** Hydrographic section AT20 (a,b,c) sampled along the $52.3°$W meridian by the research vessel Atlantis from 2012-05-01 to 2012-05-11. Temperature and salinity profiles are estimated with OSnet (e,f) at the exact location of the sampled CTDs, by interpolating the input data at those locations. Glorys12 profiles (h,i) are colocated in time and space with the CTD profiles. The SSH in a) is averaged over the sampling period of the section. The MLDs in d) is computed with the density threshold of 0.03. Salinity segments at 200m and 800m are plotted along latitude in g) with confidence interval in blue bands around the mean value for OSnet. The contours interval for the plot section are 0.5 psu from 33 to 37 psu for salinity and $2°$C from 4 to $24°$C for temperature.




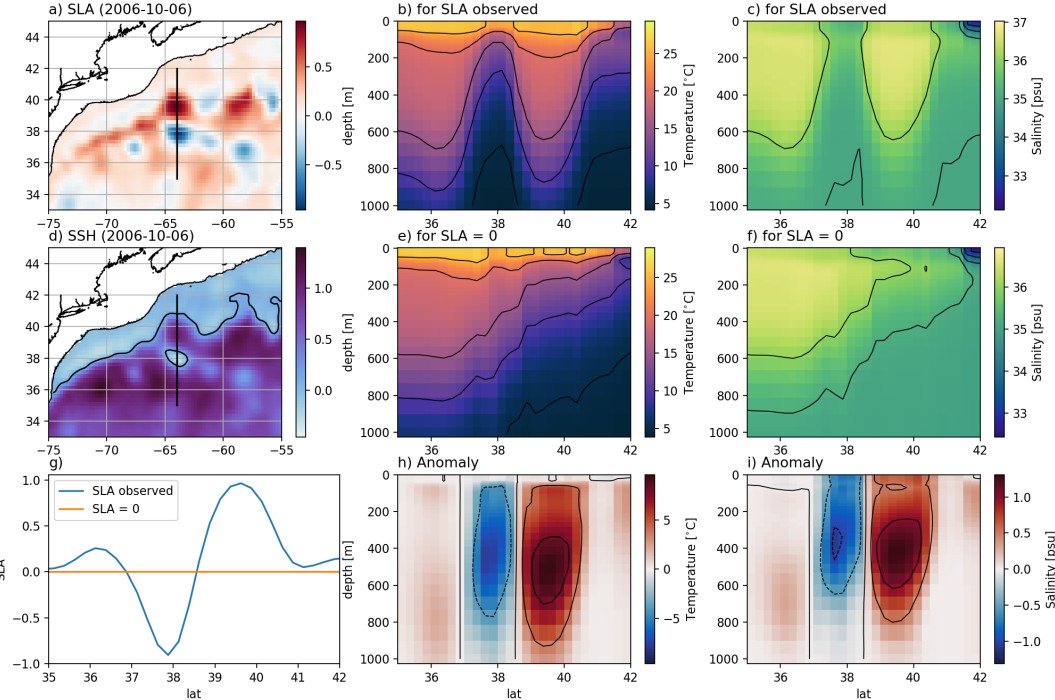

**Figure 16.** Prediction of T-S profiles for a section across two meanders of the Gulf Stream (b,c), and for a simulated Sea Level Anomaly (SLA) flattened to zero (e,f). The meanders are visualised with maps of SLA (a) and Sea Surface Height (SSH, d) which is computed by adding MDT and SLA, for the 2006-10-06. The anomaly between the T-S sections for the observed SLA and the simulated SLA is displayed in h) and i). The SSH contour in panel d) is of 0.1 m to represent the Gulf Stream.

## 5.2 OSnet to explore theoretical inputs

Since OSNet is very easy and fast to manipulate to make predictions, it could be used to make predictions using theoretical inputs. To illustrate this, we seek for the interior signature of eddies detected with altimetry. We make two predictions of temperature and salinity for a section across two particular eddies observed the $6^{th}$ of October 2006 (Fig. 16a): one prediction
is based on all observational inputs (Fig. 16b,c) and the second prediction is made by removing the eddies signature in SLA, we simply set it to zero (Fig. 16e,f,g). The interior temperature and salinity anomalies associated with the eddies are obtained by difference (Fig. 16h,i). Anomalies are the largest at depth around 400/500 m with amplitudes of a few degrees per meter of SLA anomaly. These are of reasonable amplitudes and structure for the region (Castelao, 2014) and illustrate how OSnet could easily be used to extract more knowledge than realistic T-S predictions.





### 5.3 Potential improvement of the method

Seeing the very promising results of our study, an obvious future work would be to apply OSnet on other more challenging regions, with less data or more complicated vertical structures and different dynamics. The 3D geostrophic velocities of OSnet product could be estimated using the thermal wind equation combined with surface altimeter geostrophic currents (Mulet et al., 2012). The climate change signal in the ocean's interior is difficult to observe due to a lack of *in situ* observations (Silvy et al., 2020). The global Ocean Heat Content (OHC) is generally well-constrained among ocean reanalyses, but regional differences may warrant further investigation (Palmer et al., 2017). The boundary currents are choke points of the OHC estimation accuracy, because the relatively sparse data or low resolutions of the objective analysis products are inadequate to resolve the mesoscale structures (Liang et al., 2021). We found that OSnet respects well the SST warming trend (Fig. 12a) and the mesoscale structures and it would be interesting to apply OSnet on other boundary currents and to compare the resulting OHC with previous reconstructions (e.g. Cheng et al., 2017). Other global ocean indicators such as ocean freshwater content or steric sea level could be investigated as well.

Several improvements of the method could be made in the future. One important limitation is the current NN architecture that prevents to predict profiles of different length. It constrains the analysis to range a fixed maximum depth, and forbid to keep profiles shorter than that maximum depth. A solution would be to develop a custom loss that can deal with empty variables. Another way would be to add depth as an input variable like Buongiorno Nardelli (2020) proposed. In that case, we could predict properties over shallow bathymetry and also deeper than our rather arbitrary 1000 m limit.

OSnet produces coherent horizontal and temporal patterns even though each profile is predicted independently. Yet we wonder how the horizontal and temporal surface gradients as inputs could improve predictions, especially in frontal regions. To test this, we would need to work with three dimensional (lon/lat/time) patches of input data for each profile's location and to build a different NN architecture (e.g. Ouala et al., 2018; Jouini et al., 2013; Tandeo et al., 2013; Lguensat et al., 2018) that takes patches of data as input and profiles as output. Convolutional neural architectures accounting for irregularly-sampled space-time observations might also be appealing (Fablet et al., 2021). The expected result would be sharper fronts, as we observe that the OSnet fronts are smoother than observations for the example of figure 15. We wonder if the NN could learn from the temporal surface gradient to anticipate vertical changes of stratification.

Prediction intervals (PI) also called "coverage probability" could be computed in supplement of confidence intervals (Khosravi et al., 2011). While the confidence intervals (Fig 7) gives the range of variation of a set of NN with different initializations and training datasets, the PI gives the probability for the observation associated with the prediction to be in a range of values. The PI can be obtained by adding output variables to the proposed architecture. Here, the PI could be represented as temperature and salinity profiles around the prediction, representing the 95% interval. It means there would be a 95% probability for the true value associated with the prediction to lie within the interval.



## 6   Conclusions

We proposed a method to estimate the ocean stratification from surface data using a neural network trained from *in situ* historical data. The originality of this study is the attention we give to the vertical coherence of the T-S profiles, in particular the accuracy of the MLD predictions and the absence of unrealistic vertical density inversions. The global RMSE of T and

S are better than a state of the art ocean re-analysis prediction (Glorys12's) but worst than Armor3D's predictions. However OSnet prediction do not present any unphysical density inversions while Armor3D does. Each OSnet profile is predicted independently but still produces coherent horizontal patterns on a 1/4° daily grid, especially maps of MLD. In addition, the pre-ARGO years are well reconstructed, which supports the good generalization properties of the network. Confidence intervals issued from the bootstrapped method provide an estimation of the prediction variability. The confidence is lower in the cold

surface waters north of the Gulf Stream and in the jet at depth, highlighting the most variable areas. The reconstructed surface temperature reproduces the warming trend. The seasonal cycle of surface salinity matches best the one of SSS, compared to Glorys12 and Armor3D. One convenient feature of OSnet is the possibility to estimate profiles at any location, given that the surface data is provided. This allows to compare predicted profiles at the exact location of observed CTD for example. It is computationally inexpensive to run and we encourage anyone who needs to predict ocean's stratification from surface data

to use OSnet. Another feature is the possibility to compute the relative importance of each input for each T-S prediction and analyse which surface feature influences most which property. This is a development tool that can also be used to study how the interior of the ocean's behaviour reflects on the surface data. Finally, the horizontal resolution of OSnet is constrained at a ¼° by the resolution of the SLA. The upcoming satellite mission SWOT could provide sufficient high-resolution observations for OSnet to learn from and predict smaller scale features.

*Code and data availability.*   The complete code to process input data and to develop a fully trained OSnet model is available at https://github.com/euroargodev/OSnet. A simpler version, focusing on making predictions with OSnet is also available at https://github.com/euroargodev/OSnet-GulfStream. The OSnet gridded temperature and salinity daily fields of the 0-1000m Gulf Stream region, from 1993 to 2019 are available at https://doi.org/10.5281/zenodo.6011144.

The CORA hydrographic profiles are available at https://www.seanoe.org/data/00351/46219/. MDT CNES-CLS2018 is available at https:

//www.aviso.altimetry.fr/en/data/products/auxiliary-products/mdt/mdt-global-cnes-cls18.html. SST dataset is available at https://resources.marine.copernicus.eu/product-detail/SST_GLO_SST_L4_REP_OBSERVATIONS_010_024. SLA and derived variables are available through the CMEMS portal at https://www.copernicus.eu/en/access-data/copernicus-services-catalogue/global-ocean-gridded-l4-sea-surface-heights-and-derived SSS CCI dataset is available at https://catalogue.ceda.ac.uk/uuid/4ce685bff631459fb2a30faa699f3fc5. Armor3D is available through the CMEMS portal at https://doi.org/10.48670/moi-00052. Glorys12 is available through the CMEMS portal at https://doi.org/10.48670/moi-00021.

Bathymetry ETOPO1 can be found at https://www.ngdc.noaa.gov/mgg/global/. The Line W mooring data is available at https://scienceweb.whoi.edu/linew/. The hydrographic section AT20 is accessible at https://cchdo.ucsd.edu/cruise/33AT20120419.



**Appendix A: Alternative way to suppress the density inversions with a physics-constrained loss function**

**A1    Custom loss function**

Without constraining the predictions in a physical space, most profiles show spurious density inversions that makes the MLD

computation impossible. To alleviate these issues we develop a custom loss, that constrain the density profile to be monotonous, and the properties in the mixed layer to be well mixed. The loss function is the minimisation of the mean square error between our prediction and the target profiles, that we complement with a physics-constrained loss $\text{Loss}_{\text{Phy}}$ :

$$\text{Loss} = \frac{1}{N}\sum_{n=1}^{N}(\hat{y}_n - y_n)^2 + \text{Loss}_{\text{Phy}} + \text{Loss}_{\text{H}}, \tag{A1}$$

with $N$ the batch size, $\hat{y}$ the predicted and $y$ the observed profiles of temperature and salinity as tensors of size $N \times D \times 2$ ($D = 51$ depth levels). To that standard loss we add two more terms. First we include the potential density $\sigma_0$ profile in the

target $y$ and prediction $\hat{y}$. It ensures that the T-S predictions correspond to a profile of density closer to the observed density profile. We also take the profile $K$ out of the standard loss and multiply by a coefficient $\lambda_{\text{MLD}}$ :

$$\text{Loss}_{\text{H}} = \lambda_{\text{H}}.\frac{1}{N}\sum_{n=1}^{N}(\hat{H}_n - H_n)^2 \tag{A2}$$

Second we add a constrain of monotony on the density profile to penalise the predictions that contain density inversions. A positive value of $\Delta\sigma_0$ is a violation of the hydrostatic stability of the water column. Such density inversion can exist in observed profiles at small temporal and vertical scale. As our predicted profiles are daily averages, we assume that they should

not present any density inversions, i.e. $\Delta\sigma_0 < 0$ strictly.

$$\text{Loss}_{\text{Phy}} = \lambda_{\text{Phy}}.\frac{1}{N}\sum_{n=1}^{N}\text{ReLU}\left(\frac{\Delta\sigma_0}{\Delta z}\right) \tag{A3}$$

**A2    Optimisation of the $\lambda$ coefficient**

The $\lambda$ coefficient of our custom loss needs to be optimised in order to minimize three metrics. A metric of accuracy that is the root mean square error of the target relatively to the prediction :

$$m_1 = \sqrt{\frac{1}{N}\sum_{n=1}^{N}(\hat{y}_n - y_n)^2} \tag{A4}$$

and two metrics of physical consistency, the root mean square error of the mixed layer depth H :

$$m_2 = \sqrt{\frac{1}{N}\sum_{n=1}^{N}(\hat{H}_n - H_n)^2} \tag{A5}$$

and $m_3$ the count of density inversions. Note that the $\hat{H}$ in $m_2$ is directly predicted by the NN, it is not computed on the predicted profiles with the density criterion. This is a multi objective problem that we solve with the NSGA-II genetic algorithm (Deb et al., 2002).



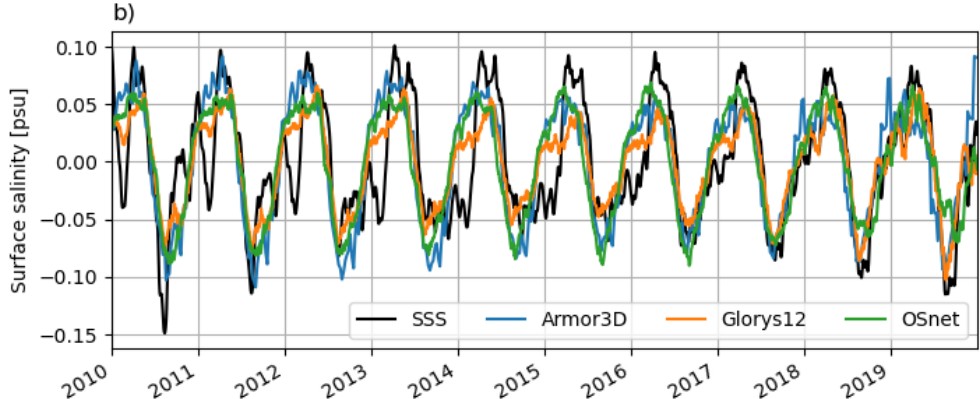

**Figure A1.** Periodic signal of the mean surface salinity from OSnet (green), Glorys12 (orange), Armor3D (blue) and remote sensing (black). The periodic signal is extracted using a STL decomposition. The SSS seasonal variation is delayed each year in winter until the 2017-2018 winter.

## Appendix B: Delay in the SSS-CCI seasonal variations

We observe a delay in the SSS seasonal variation. It is fresher by almost 0.1 psu from January to March (Fig. 13b). The periodic signal of SSS is different from 2010 to 2017, compared to the three other products but seems corrected for the 2018 and 2019 winters (Fig. A1). The authors of the SSS-CCI dataset (Boutin et al., 2021) also noticed larger seasonal biases in the SSS in respect to Argo salinities before mid-2015, over the global ocean. The largest differences relevant for our region are observed at high latitudes cold waters and boreal winter above $47°N$. After 2015 the integration of a new satellite (SMAP) and a change in the calibration mode of the satellite used over the period 2010-2019 (SMOS) in November 2014 improved the quality of the seasonal signal (Boutin et al., 2021).



*Author contributions.* GM proposed the project of using NN to predict T-S profiles and did preliminary analyses. LB and EP developed the python codes and GM tested it and wrapped it in a user friendly package. LB, PT and RF brought expert advices on neural networks, KB eased the access to datasets and helped with the general workflow, AMT, FR and GM provided ideas for the development of the method and the oceanographic pertinence of the study. EP wrote the paper and all coauthors contributed.

*Competing interests.* The authors declare no competing interests.


*Acknowledgements.* EP is funded by the Euro-Argo RISE project of the European Union's Horizon 2020 research and innovation programme under grant agreement No 824131. AMT is supported by CNRS and by the MEDLEY project,funded by JPI Climate and JPI Oceans under





the 2019 joint call. EP would like to thank Tanguy Szekely, Camille Lique, Claude Talandier and Alexandre Supply for the useful discussions around the study and Sean Tokunaga for his inspiring preliminary work on this subject.



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
