# Peer review of "Four-dimensional temperature, salinity and mixed layer depth in the Gulf Stream, reconstructed from remote sensing and in situ observations with neural networks."

_EGUsphere, 2022_

## Author Comment (AC1)

Summary:

This paper presents an approach for predicting the vertical profiles of temperature and salinity over the top 1000 m from satellite surface observations by training an empirical machine learning model using in-situ profiles in the western North Atlantic Ocean. The paper emphasises the treatment of the mixed layer depth and, specifically, a procedure to remove negative stratification from the profiles.

Overall, I found the paper to be an interesting contribution with sufficient novelty to be valuable. The ultimate impact of the work remains to be seen, but I think the paper will be worthy of publication after revision.

We thank reviewer #1 for his review. It allowed to clarify the text in several places. We changed the RMSE to a normalized RMSE that is indeed better suited. We also tried to compare OSnet to other climatologies (Roemmich and Gilson 2009, ISAS) but did not find it very instructive since there is a significant mismatch in temporal and spatial resolutions in all these products.

My major concerns are as follows:

I find it surprising and confusing that the paper does not carefully separate capability to model the 4-D climatological annual-cycle from capability to model 4-D anomalies from this climatological annual cycle. Perhaps such an approach is superior, and the methods are fine as they are, but the evaluation should clearly separate errors in the climatology from errors in the anomalies therefrom. I think the paper would be stronger if it included more explicit and quantitative evaluation of model performance on anomalies from the climatology (Nonetheless, I like the illustrative examples).

We think section 4.5 explicitly shows that OSnet captures well the surface seasonal and inter-annual variability (Fig. 12 and 13). Figure 12 is especially showing the anomalies from the climatological annual cycle (without the small frequency noise too). We only plotted the surface trends because it is comparable with SSS and SST.
Regarding the spatial anomaly relatively to the climatology : we note that in the Gulf Stream the time mean climatological field is not a relevant physical state and is never observed. A "steady" Gulf Stream is intrinsically unstable and hence never shows a laminar straight path but rather meanders and detached eddies. So the mathematical decomposition in annual cycle and anomalies is not relevant here. The eddies are an intrinsic component of the signal. No action is taken for this comment.

Relatedly, given that the method predicts the climatological annual cycle, I think the paper would be stronger if results were compared to a climatology obtained by objective mapping or optimal interpolation, e.g. updated Roemmich and Gilson 2009 gridded Argo climatology or the mean of the CORA gridded product.

—> L116 We added this explanation "We compare OSnet gridded fields to Armor3D and Glorys12 because they are the only ocean products, to our knowledge, that extend from 1993 to today with a spatial resolution of at least 1/4 of a degree and a frequency under the month (weekly for Armor3D and daily for Glorys12)."

We tried the comparison with Roemmich and Gilson 2009 Argo (RG2009) climatology but their spatial resolution of 1/2 a degree makes it hard to have the same spatial coverage than us, especially near the shelf where several cold and fresh grid points are shifting their mean value compared to ours (see the figure below with the surface temperature and salinity mean timeseries where we see the RG2009 temperature colder and salinity fresher than all other products).

More importantly, the RG2009 climatology only extends from 2004 to today, so their annual cycle would be biased towards those years. Or we could compute the annual cycle for OSnet only for

2004-2019 to validate the RG climatology which is not the scope of our paper. Similarly the ISAS gridded fields (https://www.seanoe.org/data/00412/52367/) extends from 2002 to 2020 with 1/2 degree resolution and monthly means only.

Finally we believe that if OSnet compares well with Glorys12, it has to compare well with any climatology as well, as Glorys12 is constrained by the same observations than other climatologies.

[Figure]

In the training, it seems that the selection of cross-validation data does not account for spatial and temporal autocorrelation. It is not clear that the testing data are independent of the training data. Perhaps this is ok, given that you're trying to predict or map the climatology. But, the paper would be stronger if more explicit effort was made to train and test on truly independent data (at least with regard to modelling the anomalies).

It is true that our test and train data are not independent.
—> L152 We added a mention of that caveat in the method "Be aware that the train and test data are not truly independent, the selection is random without accounting for spatial and temporal autocorrelation."
However if the *test* and *train* datasets were not completely independent, we would have exactly the same RMSE for both, but in fact there is a small difference for each model. Predictions from *test* data are always worst than those from the *train* ones.

About "you're trying to predict or map the climatology.", we believe there is a confusion here, we predict daily ocean stratification on a grid, from which we can extract averages (the climatology).

Confidence intervals or uncertainty. I'm a bit confused about how these are calculated and thus how to interpret them. The paper would be stronger if this was clearer.

The confidence intervals are the standard deviation of the 15 predictions (from the 15 models of the bootstrap). This is stated in the method L166 :"Overall, given 15 trained models, we compute the mean T, S and K profiles for each input data and their standard deviation (Fig. \ref{RMSE}, grey). The latter deliver an estimate for a confidence interval."
—> We changed the wording to "confidence interval" instead of uncertainty when it appears in the text to clarify.

—> We also added a text in the caption of the figures when the confidence intervals appear to repeat the information "[…] i.e. the standard deviation of the 15 bootstrapped models."

The main quantitative metric used is root-mean-square-error in physical units. I appreciate that this is physically intuitive, but this may obfuscate the generic statistical properties of the predictions. The paper would be stronger if normalized error metric were included, e.g. some sort of relative error and correlation.

Thank you for this proposition. We agree and changed all occurences of RMSE to a normalized RMSE. We divide the RMSE by the standard deviation of the measured property, by depth. It is a percentage that can now be compared with predictions from other regions or other dataset.

The word "coherence" is used a lot to refer to a desirable property of the 4-D gridded fields. Is this related to the frequency/waveform of the signal?? I'm not sure I understand exactly what is meant by coherence and why it is a valuable property of the predicted field. For example, in some cases, it may be that "smoothness" is unrealistic, e.g. in MLD predictions from GLORYS. Is coherence related to smoothness?

We use the words "vertical coherence"  3 times to talk about the MLD prediction accuracy and the absence of density inversions. We explain each time what we mean: e.g. " the presence of density inversions and the accuracy of the MLD prediction."
We also use the word coherence L294 to characterise the ARMOR3D fields of MLD that are often patchy (Figure 8d), it is explained in the sentence.

Be more specific about what properties of a gridded T/S dataset make it useful for interpreting local oceanographic measurements or for process studies. I'm not sure what you mean? Low error? Correlation with real variability

We want to obtain a product that is as close as possible from observations, while being physically consistent. We already introduce this goal and how it is important to have a proper MLD for climate studies (paragraph L55 to L65).
—> L357 we added a sentence to repeat this goal after the "for interpreting local oceanographic measurements or for process studies" sentence : "The goal is to be as close as possible from observation, while being physically consistent."

There are several areas where minor typographical and grammar issues need to be corrected.

We corrected several english mistakes and typos thanks to the second reviewer and our careful re-reading of the manuscript.

---

## Author Comment (AC2)

Review of the paper submitted to egusphere 2022

Four dimensional temperature, salinity and mixed layer depth in the Gulf Stream, reconstructed from remote sensing and in-situ observations with neural networks

By E. Pauthenet et al

The paper aims at providing four-dimensional temperature, salinity, and mixed layer depth in the Gulf Stream, from sea surface satellite observations (SST and altimetry). Interpolations of surface data at depth are done with a NN trained on 67767 vertical profiles. In the operational phase, satellite data are associated with vertical profiles (Temperature, Salinity, Density and MLD) through the NN. The authors also present a procedure based on density stability to improve the MLD estimation. The subject is of scientific interest due to the lack of vertical profiles in the ocean with respect to satellite surface data. The procedure presented (OSnet) seems efficient to associate sea surface satellite data with their vertical profiles. But I found the paper difficult to read and poorly structured. It can be published after the following corrections and the rewriting of some sections.

We would like to thank Michel Crepon for his careful review of our study. it helped to improve the quality of the explanation, the paper organization and some english mistakes. We especially reorganized the method paragraph 3.3 and the last section of the discussion 5.3. We also increased the resolution and resized all the figures.

Major comments

The paper is quite long and can shorten by 30%. I suspect it presents the results of Ph.d. work of an enthusiastic student who would like to present all the details of his work and has some difficulties extracting the major conclusions.

After careful consideration, we have decided to take no special action about this unspecific comment. The main reason this paper has its length is that it presents for the first time a rather complex interpolation procedure and that it provides a thorough validation of the product as it was deemed useful for future readers. The use of neural networks in oceanography is still new and requires a detailed description of the method. In the absence of any guidance from the reviewer about which presentation he thinks unnecessary, and considering that reviewer#1 did not complain about the length of the manuscript, we have not attempted a drastic reduction in size.

The readers of Ocean Sciences are physicists and most of them are not familiar with neural networks. Section 3.1 must be rewritten with care.

Agreed, we added two sentences of general introduction to MLPs, and a reference that describes well the MLP :
—> L139 "The MLP guesses the non-linear relation between inputs and outputs, through one or more hidden layers with many neurones stacked together. The learning mechanism that allows the MLP to iteratively minimize the loss function is called backpropagation."
—> L139 we added a citation to a more general review of MLP in atmospheric science "Gardner et al., 1998".
However we cannot get further into detail because this would lengthen the manuscript, which the reviewer advises us against in his first remark. MLPs are widely used in many scientific fields and the literature describing the algorithm is rich.

I recommend specifying that the use of a NN can be decomposed into two phases well separated:

- a learning phase in which the weights of the neurons are estimated from a learning data base.

- an operational phase consisting in retrieving the profiles from the satellite data (input data base)

Agreed, we modified the introduction of the method :
—> L129 "Finally, an operational phase uses the trained network and MLD adjustment to predict T, S and MLD on daily grids from the satellite data."

The learning data must be described with care: mention the origin of the profiles, which is unclear in the present form.

We completed the sentence :
—> L83 "We use the in situ temperature and salinity (T-S) vertical profiles sampled by ARGO floats and ships […]"

The input data must be justified. It appears that there is some redundancy among them: are MDTs and SLAs independent data? I do not think that geostrophic currents content added information with respect to SLA. How do you compute geostrophic current anomalies? Are they seasonal anomalies or anomalies with respect to whole observation period? Information included in SLA are also included in the geostrophic currents. These remarks are comforted by section 4.4 which shows that some variables do not play an important role and can be neglected. Section 4.4 could be suppressed if the input variables are chosen adequately in section 3 by a simple physical reasoning or by doing an EOF on the input data.

Can you comment?

Regarding MDT and SLA we added this explanation in the method :
—> L99 : "MDT is calculated by merging information from altimeter data, GRACE, and GOCE gravity field and oceanographic in situ measurements (drifting buoy velocities, hydrological profiles) (Mulet et al 2021), while SLA is only issued from altimeter data. Keeping MDT and SLA separated allows to present their respective importance in the prediction (Figure 10 and 11)."
We also added how the geostrophic currents are computed :
—> L98 "We also use geostrophic surface velocities derived from the SLA product and distributed by CNES-CLS."
Keeping the geostrophic currents as inputs gives the information of gradient at the surface that the network does not see otherwise. Even though their mean relative importance is small (Figure 10 and 11), their importance can be large for specific profils on the edge of an eddy or in a front for example. Keeping them in the analysis is not costly and slightly improves the global prediction, by correcting profiles under large surface gradients.
Section 4.4 is an important part of this paper, as neural networks are often presented as unexplainable black boxes. We have here an innovative tool to track back how the network learns.

The procedure for improving the MLD developed in section 3.3 is an important feature of this work, but it is hard to understand. Can you reformulate it in a simpler manner? How do you estimate the parameter lambda in the K estimation?

—> L177 We restructured and simplified the presentation of section 3.3
—> L194 We reworded the explanation for the calibration of the lambda parameter : ",with $\lambda = 0.57$ the value of K corresponding to the MLD. The calibration of $\lambda$ is done by a cross-validation procedure according to the estimation bias between $\hat{T}$ at sea surface and the SST value (Fig. 5). In other words $\lambda = 0.57$ allows to adjust the MLD while keeping the mean of the difference between $\hat{T}$ and SST at zero (green in Fig. 5). We expect this value to be specific to our region and of the considered NN parameterization. It would likely require a new calibration for other case-study regions."

A simpler procedure would be to apply a median filter onto the density profiles for removing the hydrostatic instability.

Yes we can smooth the density profile but it is not possible to compute back T and S from density alone.

The significance of the sentence printed in lines 199-200 is difficult to understand.

We rewrote this sentence, we hope it is clearer now. It is a description of figure 5 :
—> L207 "Note that the direct prediction of temperature at the surface (Fig. 5a, blue) is more accurate compared to SST than *in situ* observations, because OSnet learns from SST. "

I have appreciated the scientific content of appendix A which aims at removing the density inversion with a physical constrained loss function, which is an original contribution of OSnet

Thank you.

The OSnet procedure has the characteristic of a multi-entry data base. It interpolates the profiles but does not model the physical laws connecting satellite observations and the associated vertical profiles. An original procedure using hidden Markov chain, which models these physical laws has been recently developed for retrieving vertical Chl-a vertical profiles from ocean color satellite observations (Charantonis et al, 2015, Puissant et al, 2021). Can you say some words about the philosophy of these two methods, their advantages, and disadvantages?

We believe there is a confusion here, OSnet does model the link between satellite observations and the vertical profiles, similarly to Charantonis et al.. Self-organizing maps is a clustering method based on neural network that can be used for profiles retrieval too. We do not know if it would perform better or worse than our MLP. MLP are a natural method for our problem, because we don not take into account the 2D spatial information. If we used 2D patches of input data, SOM could be better suited.
—> L50 we added a citation to Charantonis et al (2015) and Gueye et al (2014), to acknowledge their work on profile predictions from the surface using SOM. Puissant et al (2021) methodology is similar to Charantonis et al (2015).

Minor comments

Most of the figures are very small. It is difficult to extract information from them. As an example, in Figure 4, it is difficult to identify the different profiles from each other.

We increased the resolution and size of all the figures. We hope it is big enough now.

Besides, the significance of the two horizontal dotted lines must be mentioned in the figure legend

The three horizontal dotted lines correspond to the MLD as written in the figure legend.

In table 2, what are the units for the T rmse, S rmse, sigma0 rmse?

It was degree Celsius, psu and kg.m3 but it is now percentages [%] as we switched to normalised RMSE, we added the unit it in the table.

In figure 5, why the density distributions of SST and SSS are so different.

Because the difference between SSS and surface salinity is too large to be impacted by the MLD adjustment. We commented on that :
—> L209 : "The salinity difference relatively to the SSS is too large for the adjustment to cause a significant issue (Fig. 5b). Still the adjusted salinity profiles with K predicted creates a fresh bias and the use of $\hat{K}^*$ corrects that (not shown). "

Section 4.2 :  horizontal maps of T and S (Figures 6, 7) are not very useful since the authors focus their interest on the four dimensional representation of these two variables. Besides the figures are very small. I suggest replacing them by vertical sections.

We think these maps are essential to appreciate the good horizontal coherence (eddies, Gulf Stream jet…), because the network only predicts profile by profile and still manage to predict consistent fields. Moreover we already show sections in figures 15 and 16. No action taken for that comment.

Section 4.5 is interesting. OSnet is able to reproduce the SST due to global change. It could be used to process ocean data in climate study contexts. But I do not understand the sentence (lines 313-314) "The long term…. Based on loess" What do you mean by loess?

Loess is an acronym of « LOcally Estimated Scatterplot Smoothing », it is used in statistics.
—> L322 We replaced it by "local regression" for clarity

Section 5.1 justifies the use of OSnet for providing T S profiles at any location. Don't be too modest! I would change line 353 as "One major feature of OSnet is the possibility…". The detection by OSnet of the big warm eddy crossing the mooring is impressive.

Thank you, we replaced "convenient" by "major" as proposed.

Some problem, the position of the mooring W3 presented in the map figuring in little cartoon at the left top of figure 14 does not correspond to the coordinates mentioned in the figure legend!

We double checked and it is all correct. the W3 mooring is at 69.11W and 38.51N like plotted in the figure.

English must be corrected by a native English-speaking person:
There are many English mistakes

Manuscript has been read carefully and corrected.

Examples:  line 91 "is shown on figure 1a"; line 2007 "is shown on figure 4"; line 299 "The figures 10 and 11…….", instead of "Figures 10 and 11……..",

We corrected the two occurrences on "on figure" to "in figure" and removed "The" in front of figure.

line 23 "the ocean's surface is observed ……." Instead of "the ocean surface has been observed………" Too many uses of the possessive case:  line 23 "the ocean surface has been observed ….. ". In modern English, possessive case is mainly dedicated to persons.

We replaced the possessive cases with a more formal writing

Data is a plural noun (singular: datum)

Corrected, thank you.

Conclusion

This paper is useful contribution to ocean data sampling. It can be published after the above corrections are done. I also suggest 30% concatenation of the text which is too long with unnecessary presentations.